# Bridge the Modality and Capability Gaps in Vision-Language Model Selection

**Chao Yi, Yu-Hang He, De-Chuan Zhan, Han-Jia Ye**[✉]
State Key Laboratory for Novel Software Technology, Nanjing University
{yic,heyh,zhandc,yehj}@lamda.nju.edu.cn

## Abstract

Vision Language Models (VLMs) excel in zero-shot image classification by pairing images with textual category names. The expanding variety of Pre-Trained VLMs enhances the likelihood of identifying a suitable VLM for specific tasks. To better reuse the VLM resource and fully leverage its potential on different zero-shot image classification tasks, a promising strategy is selecting appropriate Pre-Trained VLMs from the VLM Zoo, relying solely on the text data of the target dataset without access to the dataset's images. In this paper, we analyze two inherent challenges in assessing the ability of a VLM in this Language-Only VLM selection: the "Modality Gap"—the disparity in VLM's embeddings across two different modalities, making text a less reliable substitute for images; and the "Capability Gap"— the discrepancy between the VLM's overall ranking and its ranking for target dataset, hindering direct prediction of a model's dataset-specific performance from its general performance. We propose VLM **S**election **W**ith g**A**p **B**ridging (Swab) to mitigate the negative impact of two gaps. Swab first adopts optimal transport to capture the relevance between open-source and target datasets with a transportation matrix. It then uses this matrix to transfer useful statistics of VLMs from open-source datasets to the target dataset for bridging two gaps. By bridging two gaps to obtain better substitutes for test images, Swab can *accurately* predict the performance ranking of different VLMs on the target task *without the need for the dataset's images.* Experiments across various VLMs and image classification datasets validate Swab's effectiveness. Code is available at: https://github.com/YCaigogogo/SWAB.

## 1 Introduction

Vision-Language Models (VLMs) [46, 22, 48, 68] have demonstrated impressive image-text matching ability. One notable application of VLMs is zero-shot image classification [46, 40, 14, 37], where VLMs are leveraged to generate image classifiers using only class names directly. This zero-shot approach has shown considerable success in scenarios with scarce or no training images [35, 18].

Despite the success of VLM in image classification, the performance of a VLM may vary substantially according to the datasets and domains [11], making it challenging to use a single model to handle all tasks. Fortunately, many open-source VLMs are available [21], and these VLMs form a vast VLM Zoo. With different architectures, pre-training datasets, or training methods, these VLMs have different strengths. The diverse pre-trained VLMs increase the likelihood of pinpointing at least one VLM that excels in a given target dataset in most cases.[1] To more effectively reuse the VLM Zoo across diverse target tasks and unlock its full potential, we need a model selection method to choose suitable VLMs from the VLM Zoo for the target task. However, in scenarios such as zero-shot image

---

[1]Throughout this paper, the term "VLM" specifically refers to a *pre-trained* VLM.

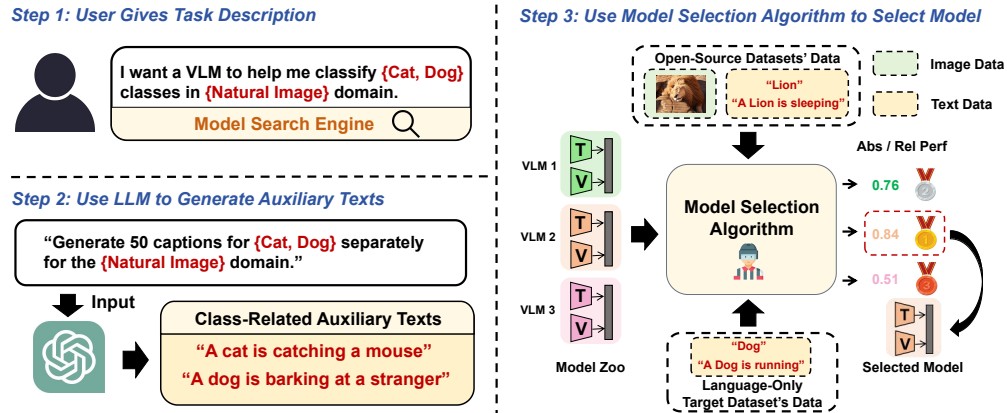

Figure 1: **Paradigm of Language-Only VLM Selection (LOVM).** Users describe the details of their target tasks in text form, such as class names and image domains. Then, LOVM utilizes this information to generate class-related labeled texts through ChatGPT. These texts serve as substitutes for image samples in subsequent model selection algorithms. The model selection algorithm uses two types of data, including the open-source datasets (which have image and text data) and the text data from the target dataset, to predict the VLM's absolute or relative performance on a target dataset. It then selects the most appropriate VLM based on the predicted performance.

classification, many users might not have labeled images for their target tasks, especially those who are not Machine Learning researchers. They prefer to describe their needs in text and use a Model Search Engine to find the most suitable model. So one solution is *identifying the most suitable VLMs in the zoo for a target dataset without access to the dataset's images*. This VLM selection is termed as "Language-Only VLM Selection" (LOVM) [73], and the paradigm is illustrated in Figure 1.

Two key types of information are available for LOVM. One is the target dataset's text data, *i.e.*, names of the target classes and class-related labeled texts generated by LLMs (Details described in Section B.1). The other is the open-source datasets, collected in the form of images with their corresponding class names. Based on these data, the goal is to estimate a VLM's zero-shot image classification capability ranking among the VLM zoo on the target dataset. LOVM encounters two challenges stemming from the inherent heterogeneity in models and datasets. The first challenge is the **Modality Gap** across different modal features extracted by a VLM. Since the visual and textual features extracted by VLMs tend to cluster into two distinct groups and have gap vectors between them [31], using text data as image proxies to rank VLMs is inaccurate. The second challenge is the **Capability Gap** between the VLM's overall ranking and its ranking in the specific dataset. Owing to the VLM's performance variation across different datasets, the VLM's average performance on open-source datasets is hard to reflect its performance on a specific target dataset. Thus, selecting a VLM based solely on its general strength may prove to be a less effective strategy.

In this paper, we propose VLM **S**election **W**ith g**A**p **B**ridging (SWAB) to address both gaps. The key idea is to reuse VLMs' statistics from open-source datasets to estimate their statistics on the target dataset, which mitigates the negative impact of these two gaps. In particular, SWAB first uses optimal transport to calculate the transport matrix based on textual similarity between class names of open-source and target datasets. After applying VLMs on open-source datasets to calculate VLMs' statistics, *i.e.*, the class-specific modality gap vectors and performance rankings of different VLMs, SWAB utilizes these statistics to estimate the same type of statistics on the target dataset. After that, SWAB uses the estimated gap vectors to align the features of texts with the features of images from the corresponding category, which bridges the modality gap. Meanwhile, SWAB's estimated VLMs' ranking also improves the prediction of their rankings on the target task, bridging the capability gap. The related work is in the appendix C. The main contributions are:

- We analyze two key challenges in LOVM — the *modality gap* across VLM's modal features and the *capability gap* between the VLM's overall ranking and its ranking on the target dataset.
- We propose SWAB, which utilizes optimal transport to transform useful statistics of VLMs on open-source datasets to the target dataset to bridge two gaps.
- Experimental results on a LOVM benchmark composed of a wide range of VLMs and image classification datasets demonstrate the effectiveness of SWAB.

## 2  Preliminary

We formally introduce the LOVM setting, a baseline method for LOVM, and analyze the two kinds of gaps in LOVM. We use $\|\cdot\|$ to represent the Euclidean norm of a vector unless otherwise defined.

### 2.1  Selecting VLMs from a Model Zoo

**Zero-Shot Image Classification of VLM.** Assume there is a pre-trained VLM $f = (f^I, f^T)$ consisting of an image encoder $f^I$ and a text encoder $f^T$. Given an image classification dataset $\mathcal{T}$ with $k_{\mathcal{T}}$ class names $C_{\mathcal{T}} = \{c_1^{\mathcal{T}}, \cdots, c_{k_{\mathcal{T}}}^{\mathcal{T}}\}$, we input the class names $C_{\mathcal{T}}$ (probably with templates like "A photo of {class}") into the VLM's text encoder $f^T$ to get the image classifiers $\{\hat{\boldsymbol{t}}_j\}_{j=1}^{k_{\mathcal{T}}}$. Then, given a test image $\boldsymbol{x}_i$, we use the image encoder $f^I$ to extract its feature $\hat{\boldsymbol{x}}_i$. Finally, we predict the label via the cosine similarity between the image feature $\hat{\boldsymbol{x}}_i$ and image classifiers $\{\hat{\boldsymbol{t}}_j\}_{j=1}^{k_{\mathcal{T}}}$. The class with the highest cosine similarity to the image is selected as the predicted class $\hat{y}_i$. Given $\hat{\boldsymbol{x}}_i = f^I(\boldsymbol{x}_i)$, $\hat{\boldsymbol{t}}_j = f^T(c_j^{\mathcal{T}})$, Equation 1 describes this zero-shot image classification process:

$$\hat{y}_i = f(\boldsymbol{x}_i, C_{\mathcal{T}}) = \underset{c_j^{\mathcal{T}} \in [C_{\mathcal{T}}]}{\operatorname{argmax}} \frac{\hat{\boldsymbol{x}}_i^{\top} \hat{\boldsymbol{t}}_j}{\|\hat{\boldsymbol{x}}_i\| \cdot \|\hat{\boldsymbol{t}}_j\|}. \tag{1}$$

**VLM Zoo.** In recent years, a large number of (pre-trained) VLMs have emerged. Assume a collection of $M$ VLMs constitute a VLM Zoo $\mathcal{M} = \left\{ f_m = (f_m^I, f_m^T) \right\}_{m=1}^{M}$. The capability of $f_m$ is determined by three key factors: the model architecture (*e.g.*, Transformer [57], ConvNeXt [34]), the pre-trained dataset (*e.g.*, LAION-400M [47], MS-COCO [32]), and the training method (*e.g.*, contrastive loss [46], caption loss [67]). Combinations of these factors result in "good and diverse" VLMs in $\mathcal{M}$. Given a dataset $\mathcal{T}$, it is probable to find a suitable VLM from the VLM zoo with high zero-shot image classification performance on $\mathcal{T}$.

**Language-Only VLM Selection (LOVM).** Rather than using images from the target dataset, LOVM focuses on the zero-shot scenario where only the target dataset's text data, such as its class names $C_{\mathcal{T}}$, are available for VLM selection. Besides, we can obtain some open-source image classification datasets $\mathcal{S}$. The set of class names in $\mathcal{S}$ is $C_{\mathcal{S}} = \{c_1^{\mathcal{S}}, \cdots, c_{k_{\mathcal{S}}}^{\mathcal{S}}\}$, and the $D_{\mathcal{S}}^I$ denote the labelled images in these classes. Given a target task $\mathcal{T}$, the VLM selection method $h$ estimates the zero-shot classification ability of $f_m$ based on $C_{\mathcal{T}}$, $C_{\mathcal{S}}$, and $D_{\mathcal{S}}^I$ via $\hat{r}_m^{\mathcal{T}} = h(f_m \mid C_{\mathcal{T}}, C_{\mathcal{S}}, D_{\mathcal{S}}^I)$, where $m \in [1, \cdots, M]$. $\hat{r}_m^{\mathcal{T}}$ is the predicted ranking of the $m$-th VLM $f_m$ on $\mathcal{T}$. The higher the ranking, the more probable $f_m$ achieves higher zero-shot image classification performance on $\mathcal{T}$. Assuming we can obtain the test image set $D_{\mathcal{T}}^I$ of the target dataset $\mathcal{T}$ with $|D_{\mathcal{T}}^I|$ images, then we can calculate the zero-shot image classification accuracy $p_m^{\mathcal{T}}$ of $f_m$ is calculated by $p_m^{\mathcal{T}} = \frac{1}{|D_{\mathcal{T}}^I|} \sum_{(\boldsymbol{x}_i, y_i) \in D_{\mathcal{T}}^I} \mathbb{I}(y_i = f_m(\boldsymbol{x}_i, C_{\mathcal{T}}))$. $f_m(\boldsymbol{x}_i, C_{\mathcal{T}})$ represents the predicted class with the same manner as Equation 1. $\mathbb{I}(\cdot)$ is the indicator function, which outputs 1 if the condition is satisfied, and 0 otherwise. Based on $\{p_m^{\mathcal{T}}\}_{m=1}^{M}$, we obtain the true ranking of $M$ VLMs $\boldsymbol{r}^{\mathcal{T}} = [r_1^{\mathcal{T}}, \ldots, r_M^{\mathcal{T}}]$ by assigning higher ranking $r$ to models with higher accuracy $p$. However, in the zero-shot scenario, we can't obtain the test images set $D_{\mathcal{T}}^I$ in advance. Therefore, the goal of LOVM is to make the predicted ranking $\hat{\boldsymbol{r}}^{\mathcal{T}} = [\hat{r}_1^{\mathcal{T}}, \cdots, \hat{r}_M^{\mathcal{T}}]$ be an accurate estimation of the ground truth ranking $\boldsymbol{r}^{\mathcal{T}} = [r_1^{\mathcal{T}}, \ldots, r_M^{\mathcal{T}}]$ so that the best VLM can be selected.

**Evaluation of LOVM Methods.** We measure the performance of the LOVM algorithm by comparing the ranking similarity between $\boldsymbol{r}^{\mathcal{T}}$ and $\hat{\boldsymbol{r}}^{\mathcal{T}}$. Specifically, we calculate the Top-5 Recall $R_5$ (ranges from 0 to 1) and Kendall's Rank Correlation $\tau$ (ranges from -1 to 1). The larger the better.

### 2.2  Possible Paradigms for LOVM

**Non-Learning-based LOVM.** There are three main paradigms for LOVM. *The first paradigm* is to neglect the visual encoder and select VLM solely on texts. In detail, we can utilize ChatGPT [43] to generate auxiliary texts $\tilde{D}_{\mathcal{T}}$ based on class names $C_{\mathcal{T}}$ of $\mathcal{T}$. More details are described in Section B.1. These class-specific texts act as "image proxies". Then, whether a VLM $f_m$ fits $\mathcal{T}$ could be measured by transferability metrics, *e.g.*, H-Score [2] and LogME [65], between the VLM's text encoder $f_m^T$ and generated text dataset $\tilde{D}_{\mathcal{T}}$. *The second paradigm* relies on the general performance

of a certain VLM $f_m$. We use open-source datasets to measure a VLM's general performance. If $f_m$ achieves high zero-shot classification performance over open-source datasets, then it is expected to be competitive on $\mathcal{T}$. These methods assume that a VLM's ranking is relatively consistent across tasks.

**Learning-based LOVM**. *The third paradigm* is based on the learning process. In detail, the ability of a VLM could be predicted based on a ranker model $f_R$. The input of $f_R$ is a vector $s_m^{\mathcal{T}}$, depicting the dataset-specific representation of $f_m$ on $\mathcal{T}$, while the output of $f_R$ is the relative/absolute performance $\hat{p}_m^{\mathcal{T}} \in \mathbb{R}$ of $f_m$ on $\mathcal{T}$. The $f_R$ could be *learned* on open-source datasets $\mathcal{S}$ [70, 73]. Due to the availability of both class names $C_{\mathcal{S}}$ and images $D_{\mathcal{S}}^I$ in the open-source dataset $\mathcal{S}$ such as ImageNet [8], we can calculate each VLM's representation $\{s_m^n\}_{m=1,n=1}^{M,N}$ and true zero-shot image classification accuracy $\{p_m^n\}_{m=1,n=1}^{M,N}$. Here $N$ refers to the number of datasets in $\mathcal{S}$. After constructing the train set, the ranker model $f_R$ is learned based on the $\{s_m^n, p_m^n\}_{m=1,n=1}^{M,N}$:

$$\min_{f_R} \sum_{m=1}^{M} \sum_{n=1}^{N} \ell(f_R(s_m^n), p_m^n). \tag{2}$$

$\ell$ is a loss function that measures the discrepancy between the prediction and the ground truth, which can be Mean Squared Error Loss and Huber Loss, among others. Given $\mathcal{T}$, the learned $f_R$ is able to predict the performance $\{\hat{p}_m^{\mathcal{T}}\}_{m=1}^{M}$ over $\{s_m^{\mathcal{T}}\}_{m=1}^{M}$ via $\hat{p}_m^{\mathcal{T}} = f_R(s_m^{\mathcal{T}})$. Finally, we can get the predicted VLMs' ranking $\hat{r}$ based on $\{\hat{p}_m^{\mathcal{T}}\}_{m=1}^{M}$. This approach has similarities with meta-learning [12, 3]. Meta-learning attempts to use data from multiple datasets to learn a model adaptable to the target task, while Learning-based LOVM employs data from multiple datasets to learn a ranker model for selecting the suitable model from a VLM Zoo for the target task. The representation $s_m^{\mathcal{T}}$ is one of the keys in this paradigm, and ModelGPT [73] calculates values $s_m^{\mathcal{T}}$ via the capability of a VLM's text encoder $f_m^T$.

**ModelGPT** uses generated text data $\tilde{D}_{\mathcal{T}}$ as substitutes for images to calculate some metrics, which measures the zero-shot ability of $f_m$ on unseen images by the classification ability of $f_m$ on $\tilde{D}_{\mathcal{T}}$:

$$s_{m,i}^{\mathcal{T}} = \text{Metric}_i\left(f_m, \tilde{D}_{\mathcal{T}}\right). \tag{3}$$

Here $\text{Metric}_i$ indicates the $i$-th metrics function such as Top-1 Accuracy and F1-Score. For example, the Top-1 Accuracy $s_{m,1}^{\mathcal{T}}$ could be calculated in a similar manner as Equation 1, with the only difference being that the test samples were replaced with text samples $t_i$ instead of image samples $x_i$:

$$s_{m,1}^{\mathcal{T}} = \frac{1}{|\tilde{D}_{\mathcal{T}}|} \sum_{(t_i,y_i)\in\tilde{D}_{\mathcal{T}}} \mathbb{I}\left(y_i = f_m(t_i, C_{\mathcal{T}})\right). \tag{4}$$

Besides, ModelGPT uses some metrics for assessing the features' quality extracted by the VLM's text encoder $f_m^T$. More details are in the Section B.2. Moreover, the zero-shot classification performance of $f_m$ on ImageNet is also included in $s_m^{\mathcal{T}}$ as a general ability measure of $f_m$. ModelGPT implements $f_R$ as a simple linear model.

## 2.3 Analysis of the Two Gaps in LOVM

There are two main challenges that limit the application of the aforementioned paradigms in LOVM. The first is the modality gap across different modalities' features in VLM's feature space, and the second is the capability gap between VLM's overall performance and dataset-specific performance. **Modality Gap.** As described in Section 2.2, methods like H-Score, LogME, and ModelGPT utilize the ChatGPT generated auxiliary texts $\tilde{D}_{\mathcal{T}}$ as image proxies to calculate metrics that measure the zero-shot accuracy on the target dataset $\mathcal{T}$. In other words, the zero-shot classification ability across text and image modalities is estimated by the intra-modality classification ability. The latent assumption is that the generated texts and their corresponding images are closely aligned in VLM's feature space. However, this assumption is difficult to meet [31], and instances' features are more likely to cluster according to their modalities. In particular, we define the modality gap vector $g$ between the features of an image-text pair $(x_i, t_i)$ as $g_{m,i} := f_m^I(x_i) - f_m^T(t_i)$. Values in the gap vector are generally not close to zero. We name this phenomenon as *Modality Gap* in LOVM, which makes the scores on $\tilde{D}_{\mathcal{T}}$ hard to reveal the true zero-shot image classification capability of a VLM on a given dataset.

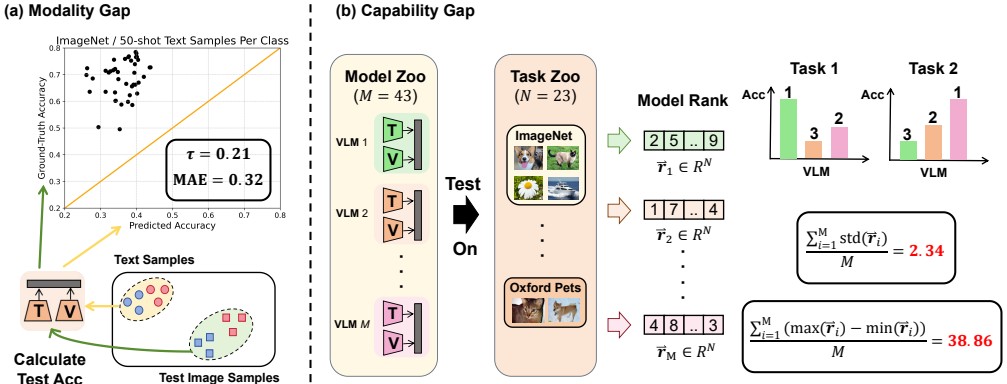

Figure 2: **Validation Experiments on the Modality Gap and Capability Gap.** (a) Predicted VLMs' zero-shot image classification accuracy based on generated text data vs. VLM's true accuracy based on test images. Each point in the graph represents a model. From the result, we can find that the predicted accuracy poorly aligns with the true accuracy, indicating these text data are ineffective substitutes for image data. (b) We calculate the zero-shot image classification performance rankings of 43 VLMs across 23 datasets. We compute the average standard deviations and the mean value of differences between each VLM's maximum and minimum ranking. The result shows the performance of a VLM varies greatly across different datasets.

We conduct a validation experiment on ImageNet with 43 VLMs. We first generate 50 auxiliary texts per class as $\tilde{D}_{\mathcal{T}}$ and then calculate the predicted Top-1 accuracy via Equation 4. Next, we use test images to calculate the VLM's true Top-1 accuracy. The consistency between the predicted Top-1 accuracy and true zero-shot image classification accuracy $p_{m,\mathcal{T}}$ is measured by the Kendall Rank Correlation ($\tau$, higher is better) and Mean Absolute Error (MAE, lower is better). It can be observed from the left part of Figure 2 that the predicted accuracy derived from auxiliary texts $\tilde{D}_{\mathcal{T}}$ does not closely match the true accuracy, indicating that these generated auxiliary texts in $\tilde{D}_{\mathcal{T}}$ are not effective proxies for images.

To make the auxiliary texts act as better image proxies, one intuitive idea is to estimate the gap vector $\boldsymbol{g}$ for each image-text pair. Then we can add it to the feature $f_m^T(t_i)$ of the text $t_i$ to eliminate the modality gap, which may lead to more accurate scores $s_{m,i}^{\mathcal{T}}$ in Equation 3. However, the gap vector cannot be calculated directly without the target dataset's images. Furthermore, gap vectors for different classes are diverse, so using a shared vector across all datasets may not be a good choice.

**Capability Gap.** To select one VLM from the model zoo given a target dataset, one direct approach is to select the VLM that performs the best on average across multiple datasets. For example, we may first estimate the VLM's zero-shot classification ability on open-source datasets and then select the VLM with the highest performance. The key question is whether a VLM's average ranking on the open-source datasets can reveal its true ranking on the target dataset. Our empirical analyses indicate that there exists a discrepancy between the VLM's overall ranking and its ranking on a specific dataset. We name the discrepancy between the VLM's average ability and its specific ability as the *Capability Gap*, which results from the fact that a VLM's performance fluctuates significantly across various datasets.

To verify the claim, we test 43 VLMs on 23 target datasets provided by [73] and obtain the rankings of each VLM across these datasets. Based on these ranking results, we calculate the average standard deviation and the mean value of the difference between each VLM's maximum and minimum ranking. The experiment process is illustrated in the right part of Figure 2. We find that the mean difference between one VLM's maximum and the minimum ranking is 38.86. Since the total number of VLMs is 43, such a difference demonstrates that the top-performing VLM in one dataset could likely be among the worst in another.

One solution to bridge such a capability gap is to consider the VLMs' ranking on a related dataset. In other words, the ranking of VLMs on datasets from open-source datasets collections that are relevant to the target task may provide more useful insights than a general performance ranking across all tasks. The main challenge is to figure out which open-source datasets are similar to the target dataset and transform the VLM's ranking on these datasets to the target dataset.

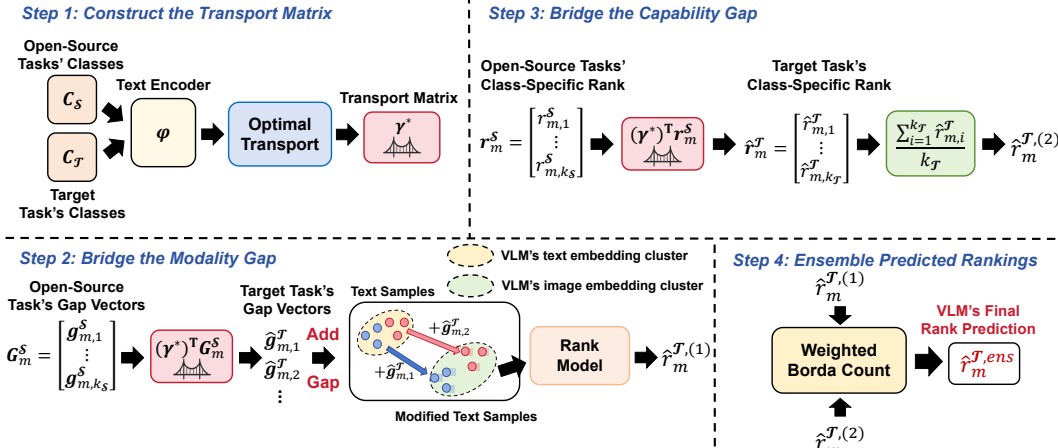

Figure 3: **The workflow of SWAB**. SWAB first constructs a transport matrix $\gamma^* \in \mathbb{R}^{k_\mathcal{S} \times k_\mathcal{T}}$ using optimal transport, based on textual semantic similarity between classes in the open-source datasets $C_\mathcal{S} = \{c_1^\mathcal{S}, \cdots, c_{k_\mathcal{S}}^\mathcal{S}\}$ and the target dataset's classes $C_\mathcal{T} = \{c_1^\mathcal{T}, \cdots, c_{k_\mathcal{T}}^\mathcal{T}\}$. Using this matrix, SWAB estimates VLM $f_m$'s class-specific gap vectors $\{\hat{\boldsymbol{g}}_{m,1}^\mathcal{T}, \cdots\}$ on the target dataset $\mathcal{T}$ from the gap vectors $\boldsymbol{G}_m^\mathcal{S} \in \mathbb{R}^{k_\mathcal{S} \times d}$ in the open-source datasets. These estimated gap vectors help modify text data to act as more effective substitutes for image data. The modified text data will then be input into the Ranker Model $f_R$, which predicts VLM's performance $\hat{r}_m^{\mathcal{T},(1)}$ on the target dataset. Besides, SWAB also uses the transport matrix $\gamma^*$ to predict VLM's performance ranking on the target dataset based on VLM's class-specific rankings $\boldsymbol{r}_m^\mathcal{S} \in \mathbb{R}^{k_\mathcal{S}}$ on open-source datasets. Finally, SWAB combines these two ranking predictions $\hat{r}_m^{\mathcal{T},(1)}$ and $\hat{r}_m^{\mathcal{T},(2)}$ to determine the VLM's final ranking prediction.

**Summary.** We emphasize two kinds of gaps in LOVM, *i.e.*, the *modality gap* across features of different modalities generated by a VLM, and the *capability gap* between a VLM's overall ranking and its ranking given a specific target dataset. Both two gaps pose obstacles to previous model selection methods, such as LogME and ModelGPT, and degrade their abilities in VLM selection. Moreover, those intuitive approaches to bridge the gaps still face challenges.

## 3 VLM Selection with Gap Bridging

To mitigate the impact of both gaps on LOVM and integrate non-learning-based and learning-based LOVM methods, we propose VLM **S**election **W**ith g**A**p **B**ridging (SWAB). The key idea of SWAB is to bridge modality and capability gaps by utilizing class-level statistics of VLMs from open-source datasets. By measuring the textual similarity between the target dataset's class names and those in open-source datasets, we construct a bridge matrix. Based on it, we estimate the gap vectors between image and text modalities, which rectifies the text-derived scores in ModelGPT. In addition, we predict the VLM's performance ranking for the target dataset based on the bridge matrix and VLM's ranking on the open-source dataset. Both estimated statistics will be used to obtain a more accurate language-only VLM selection. The workflow of SWAB is illustrated in Figure 3.

### 3.1 Construct the Bridge Matrix Using Optimal Transport

Benefiting from the open-source datasets, some useful class-level statistics, such as modality gap vectors and zero-shot classification accuracy of a certain VLM, could be calculated, which can help the performance ranking estimation of a VLM on the target dataset. To better utilize these class-level statistics for predicting the corresponding statistics of a VLM on the target task, we introduce semantic relevance information between open-source datasets' classes and target dataset's classes into the statistics reusing process, which is automatically generated through Optimal Transport [7, 45].

Recall that the sets of class names of the open-source datasets and the target dataset are $C_\mathcal{S} = \{c_i^\mathcal{S}\}_{i=1}^{k_\mathcal{S}}$ and $C_\mathcal{T} = \{c_i^\mathcal{T}\}_{i=1}^{k_\mathcal{T}}$, respectively. The semantic relevance between two classes could be measured by the textual similarity between their class names. In detail, we use a pre-trained text encoder $\phi$ (*e.g.*, MPNet [49]), which extracts text features for these class names, *i.e.*, $\{\phi(c_1^\mathcal{S}), \cdots, \phi(c_{k_\mathcal{S}}^\mathcal{S})\}$

and $\{\phi(c_1^{\mathcal{T}}), \cdots, \phi(c_{k_{\mathcal{T}}}^{\mathcal{T}})\}$. Then, we can calculate the cosine similarity $\frac{\phi(c_i^{\mathcal{S}})^{\top}\phi(c_j^{\mathcal{T}})}{\|\phi(c_i^{\mathcal{S}})\|\cdot\|\phi(c_j^{\mathcal{T}})\|}$ between the text feature of the $i$-th class in the open-source datasets $\phi(c_i^{\mathcal{S}})$ and that of the $j$-th class in the target dataset $\phi(c_j^{\mathcal{T}})$. The larger the cosine similarity, the more similar the two classes are. After that, we construct the cost matrix in optimal transport via $\text{cost}_{ij} = 1 - \frac{\phi(c_i^{\mathcal{S}})^{\top}\phi(c_j^{\mathcal{T}})}{\|\phi(c_i^{\mathcal{S}})\|\cdot\|\phi(c_j^{\mathcal{T}})\|}$. In practice, we exponentiate each element in $\text{cost} \in \mathbb{R}_{\geq 0}^{k_{\mathcal{S}} \times k_{\mathcal{T}}}$ using the base $e$ to amplify its differences. We then solve the optimal transport problem with the constructed cost matrix to get the transport matrix $\boldsymbol{\gamma}^*$. Since optimal transport aims to obtain a transport matrix that minimizes the transmission cost, the transport matrix $\boldsymbol{\gamma}^*$ will reuse more information between semantically similar classes:

$$\boldsymbol{\gamma}^* = \underset{\boldsymbol{\gamma} \in \mathbb{R}_{\geq 0}^{k_{\mathcal{S}} \times k_{\mathcal{T}}}}{\operatorname{argmin}} \sum_{i,j} \gamma_{i,j}\, \text{cost}_{i,j} \ , \ \text{s.t. } \boldsymbol{\gamma}\mathbf{1} = \boldsymbol{u}; \ \boldsymbol{\gamma}^T\mathbf{1} = \boldsymbol{v}; \ \gamma_{i,j} \geq 0. \tag{5}$$

The cost matrix quantifies the expense of moving elements between all class pairs, and $\boldsymbol{\gamma}^* \in \mathbb{R}^{k_{\mathcal{S}} \times k_{\mathcal{T}}}$ is the transport matrix. OT minimizes the cost indicated by the matrix $\text{cost}$ and moves elements from one distribution $\boldsymbol{u}$ to another $\boldsymbol{v}$. In SWAB, we define $\boldsymbol{u}$ and $\boldsymbol{v}$ as uniformly distributed vector $\boldsymbol{u} = \mathbf{1}/k_{\mathcal{S}} \in \mathbb{R}^{k_{\mathcal{S}}}$ and $\boldsymbol{v} = \mathbf{1}/k_{\mathcal{T}} \in \mathbb{R}^{k_{\mathcal{T}}}$. This indicates that we treat all classes as equally important. We may also incorporate prior knowledge of class importance to define $\boldsymbol{u}$ and $\boldsymbol{v}$.

The solution $\boldsymbol{\gamma}^*$ of the OT problem in Equation 5 could be solved efficiently [13], and $\boldsymbol{\gamma}^*$ acts as a bridge matrix between open-source datasets' classes and target dataset's classes. Usually, the smaller $\text{cost}_{i,j}$ is, the larger the corresponding element $\gamma_{i,j}^*$ obtained by OT, indicating statistics of the $i$-th class of open-source datasets may help more when we estimate the statistics of the $j$-th target class.

### 3.2 Bridge the Modality Gap and Capability Gap

**Bridge the Modality Gap.** Given the $m$-th VLM $f_m$ in the model zoo, we want to estimate the modality gap $\boldsymbol{g}_{m,j}^{\mathcal{T}}$ between the extracted image and text features for the $j$-th class in the target dataset $\mathcal{T}$ to bridge the modality gap. However, in the zero-shot scenario, we can't get the target dataset's images in advance, so we can't directly calculate the gap vectors using image-text pairs. To solve this problem, SWAB estimates the target dataset's gap vectors based on the open-source datasets' gap vectors with $\boldsymbol{\gamma}^*$. Given the $k$-th open-source class $c_k^{\mathcal{S}}$, we can get the set of images $D_{\mathcal{S}_k}^I = \left\{ (\boldsymbol{x}_i, y_i) \,|\, (\boldsymbol{x}_i, y_i) \in D_{\mathcal{S}}^I, \ y_i = c_k^{\mathcal{S}} \right\}$ from the open-source datasets. $|D_{\mathcal{S}_k}^I|$ is the number of images in $D_{\mathcal{S}_k}^I$. Then, the modality gap vector $\boldsymbol{g}_{m,k}^{\mathcal{S}}$ for class $c_k^{\mathcal{S}}$ and model $f_m$ can be calculated through $\boldsymbol{g}_{m,k}^{\mathcal{S}} = \frac{1}{|D_{\mathcal{S}_k}^I|} \sum_{(\boldsymbol{x}_i, y_i) \in D_{\mathcal{S}_k}^I} \left( \frac{f_m^I(\boldsymbol{x}_i)}{\|f_m^I(\boldsymbol{x}_i)\|} - \frac{f_m^T(c_k^{\mathcal{S}})}{\|f_m^T(c_k^{\mathcal{S}})\|} \right)$. $\boldsymbol{g}_{m,k}^{\mathcal{S}}$ is the average difference between the normalized class text prototype embedding and all normalized image embeddings from the class $c_k^{\mathcal{S}}$. In a similar manner, the gap vectors of all open-source classes $\{\boldsymbol{g}_{m,1}^{\mathcal{S}}, \cdots, \boldsymbol{g}_{m,k_{\mathcal{S}}}^{\mathcal{S}}\}$ can be obtained given $f_m$. We use a matrix $\boldsymbol{G}_m^{\mathcal{S}} \in \mathbb{R}^{k_{\mathcal{S}} \times d_m}$ to represent those $k_{\mathcal{S}}$ gap vectors for the $m$-th VLM in the VLM Zoo, and $d_m$ is the dimensionality of features extracted by $f_m$.

The gap vectors $\{\boldsymbol{g}_{m,1}^{\mathcal{T}}, \cdots, \boldsymbol{g}_{m,k_{\mathcal{T}}}^{\mathcal{T}}\}$ for the target dataset could be estimated based on the $\boldsymbol{G}_m^{\mathcal{S}}$ and the transport matrix $\boldsymbol{\gamma}^*$. If two classes are semantically similar, then we can reuse the gap vector from the similar class. We set the predicted gap vector for the $j$-th target class $\hat{\boldsymbol{g}}_{m,j}^{\mathcal{T}}$ as a weighted sum of $\boldsymbol{G}_m^{\mathcal{S}}$, and the weight comes from $\boldsymbol{\gamma}^*$, which is $\hat{\boldsymbol{g}}_{m,j}^{\mathcal{T}} = |C_{\mathcal{T}}|(\boldsymbol{\gamma}_{:,j}^*)^{\top}\boldsymbol{G}_m^{\mathcal{S}}$. $\boldsymbol{\gamma}_{:,j}^*$ is the $j$-th column of $\boldsymbol{\gamma}^*$. We use scaling factors $|C_{\mathcal{T}}|$ to ensure that for each target class, the sum of $\boldsymbol{\gamma}_{:,j}^*$ equals 1. This scale operation has also been used in previous work [62, 63]. After that, we modify the step of ModelGPT in Equation 3, where the metrics over the generated auxiliary texts $\tilde{D}_{\mathcal{T}}$ are calculated. We add the gap vector $\hat{\boldsymbol{g}}_{m,j}^{\mathcal{T}}$ to the embeddings of the auxiliary texts $\tilde{D}_{\mathcal{T}}$ from the $j$-th class in the target dataset:

$$\tilde{\boldsymbol{t}}_{m,i} = f_m^T(t_i) + \hat{\boldsymbol{g}}_{m,j}^{\mathcal{T}} \ , \ \ \forall t_i \in \tilde{D}_{\mathcal{T}}^j \ . \tag{6}$$

The modified text embedding $\tilde{\boldsymbol{t}}_{m,i}$ serves as better image proxies. In other words, classification metrics on $f_m^T(t_i)$ only reveal the discerning ability of the text encoder of $f_m$, which is far from the (cross-modal) zero-shot classification ability due to the modality gap. By bridging such a gap with modified text embedding, classification metrics on $\tilde{\boldsymbol{t}}_{m,i}$ *are closer to the classification metrics on images with textual classifier*. Therefore, we use $\{\tilde{\boldsymbol{t}}_{m,1}, \cdots\}$ in Equation 6 as better

inputs to calculate $s_m^{\mathcal{T}}$ in Equation 3. The updated score vectors $s_m^{\mathcal{T}}$ are then input to the ranker model $f_R$, which is able to get more accurate VLM's performance prediction $\hat{p}_m^{\mathcal{T}}$. Based on the performance prediction $\{\hat{p}_m^{\mathcal{T}}\}_{m=1}^M$, we can obtain the VLMs' ranking $\hat{\boldsymbol{r}}^{\mathcal{T},(1)} = [\hat{r}_1^{\mathcal{T},(1)}, \cdots, \hat{r}_M^{\mathcal{T},(1)}] =$ Ranking $\left([\hat{p}_1^{\mathcal{T}}, \cdots, p_M^{\mathcal{T}}]\right)$. Ranking$(\cdot)$ transforms the accuracies into models' ranking.

**Bridge the Capability Gap.** Whether the $m$-th VLM $f_m$ fits the target task $\mathcal{T}$ could also be estimated by the performance of $f_m$ on the open-source datasets related to $\mathcal{T}$. We first calculate the VLM's class-level performance ranking over the whole open-source datasets. Given the $k$-th open-source class $c_k^{\mathcal{S}}$ and the corresponding set of images $D_{\mathcal{S}_k}^I$, we calculate the zero-shot classification accuracy $p_{m,k}^{\mathcal{S}}$ via Equation 1. We then determine each VLM's ranking on the $k$-th open-source class $c_k^{\mathcal{S}}$ using Equation 7. We calculate VLMs' ranking for other open-source classes in the same way.

$$\boldsymbol{r}_k^{\mathcal{S}} = [r_{1,k}^{\mathcal{S}}, \cdots, r_{M,k}^{\mathcal{S}}] = \text{Ranking}\left([p_{1,k}^{\mathcal{S}}, \cdots, p_{M,k}^{\mathcal{S}}]\right) \ . \tag{7}$$

Next, we estimate the ranking of a certain VLM $f_m$ on the target dataset $\mathcal{T}$. By re-organizing the ranking values in Equation 7, the performance ranking vector of $f_m$ on all $k_{\mathcal{S}}$ open-source classes is $\boldsymbol{r}_m^{\mathcal{S}} = [r_{m,1}^{\mathcal{S}}, \cdots, r_{m,k_{\mathcal{S}}}^{\mathcal{S}}] \in \mathbb{Z}_+^{k_{\mathcal{S}}}$. If $f_m$ ranks high on certain open-source classes related to the classes in the target dataset, the performance ranking of $f_m$ on the target dataset may also be high. Thus, we perform a weighted sum of ranking values in $\boldsymbol{r}_m^{\mathcal{S}}$ and assign larger weights to those classes related to the target dataset. This can be done by using $\boldsymbol{\gamma}^*$:

$$\hat{\boldsymbol{r}}_m^{\mathcal{T}} = \boldsymbol{r}_m^{\mathcal{S}} \, \boldsymbol{\gamma}^* \ . \tag{8}$$

Elements in $\hat{\boldsymbol{r}}_m^{\mathcal{T}} \in \mathbb{R}^{k_{\mathcal{T}}}$ are the predicted ranking of the m-th VLM $f_m$ for $k_{\mathcal{T}}$ target classes. Since we only need the relative order of ranks, there is no additional scale factor in Equation 8. After that, we average class-specific ranking values in $\hat{\boldsymbol{r}}_m^{\mathcal{T}}$, and use $\hat{r}_m^{\mathcal{T},(2)} = \text{Mean}\left(\hat{\boldsymbol{r}}_m^{\mathcal{T}}\right)$ to denote the estimated ranking of $f_m$ on the target dataset $\mathcal{T}$. In summary, we predict the ranking of models on each class in the target dataset based on their rankings in classes of the open-source datasets, guided by the semantic relevance between the classes. We predict the ranking of models on each category in the target dataset based on their rankings in categories of the open-source dataset, guided by the semantic relevance between the categories. Benefiting from the models' consistently aligned classification performance ranking across similar classes, this approach bridges the capability gap in VLM selection.

### 3.3 Summary of SWAB

As is described in Section 3.2, given a target dataset $\mathcal{T}$ and a VLM $f_m$, we denote the two performance ranking predictions obtained by bridging the modality gap and capability gap as $\hat{r}_m^{\mathcal{T},(1)}$ and $\hat{r}_m^{\mathcal{T},(2)}$, respectively. $\hat{r}_m^{\mathcal{T},(1)}$ is predicted based on ModelGPT with our modified embeddings of the generated auxiliary texts for $\mathcal{T}$. $\hat{r}_m^{\mathcal{T},(2)}$ is predicted by the weighted sum of VLM's class-specific ranking values on the open-source datasets. These two predictions, respectively, originate from learning-based and non-learning-based LOVM methods. We ensemble two predictions together and achieve a more accurate model ranking estimation. We utilize the weighted Borda Count to aggregate two rankings:

$$\hat{r}_m^{\mathcal{T},\text{ens}} = \alpha \cdot \hat{r}_m^{\mathcal{T},(1)} + (1-\alpha) \cdot \hat{r}_m^{\mathcal{T},(2)}. \tag{9}$$

We set $\alpha = 0.5$. Ultimately, SWAB determines the final predicted ranking of the VLMs in the VLM Zoo based on $\hat{r}_m^{\mathcal{T},\text{ens}}$. The pseudo-code of SWAB is listed in Algorithm 1.

## 4 Experiments

### 4.1 Evaluation on LOVM Benchmark

**Setups.** We follow LOVM [73] to use its provided VLM Zoo. In addition, to further enhance the diversity of the VLM Zoo, we add some representative VLMs such as BLIP [28] and BEiT-3 [59] to expand the VLM Zoo. The final VLM Zoo contains a total of 43 models, which differ in aspects such as model architecture, pre-training datasets, and training methods. In the appendix, we also provide experimental results of different model selection methods on the 35 models originally offered by LOVM. We evaluate different methods on 23 datasets, *i.e.* ImageNet [8], Aircraft [36],

Table 1: **Results on LOVM Benchmark.** We evaluate our method across 23 datasets and 43 pre-trained VLMs. The results are averaged over all datasets. Our SWAB achieves the best results across all metrics. For methods that involve adding random noise to data features, we report the standard deviation of metrics across 10 experiments to mitigate the impact of randomness on result reliability.

| Methods | H-Score | NCE | LEEP | LogME | INB | Avg Rank | ModelGPT | SWAB |
|---|---|---|---|---|---|---|---|---|
| $R_5(\uparrow)$ | 0.174 | 0.235 | 0.161 | 0.191 | 0.443 | 0.443 | $0.446_{\pm 0.004}$ | $\mathbf{0.504}_{\pm 0.000}$ |
| $\tau(\uparrow)$ | 0.000 | -0.014 | 0.014 | -0.014 | 0.267 | 0.246 | $0.270_{\pm 0.009}$ | $\mathbf{0.318}_{\pm 0.002}$ |
| $R_5 + \tau(\uparrow)$ | 0.174 | 0.221 | 0.175 | 0.177 | 0.710 | 0.689 | $0.716_{\pm 0.011}$ | $\mathbf{0.822}_{\pm 0.002}$ |

CIFAR100 [26] and so on. We obtain VLM's ground truth ranking based on VLM's Top-1 Accuracy calculated on the target dataset's test image set.

**Baseline.** We select representative methods for each of the three paradigms mentioned in Section 2.2 as our baselines. For the first paradigm, we use four classic model selection methods: H-Score [2], NCE [55], LEEP [56] and LogME [65]. For the second paradigm, we use the VLM's ranking on ImageNet (INB) and VLM's average ranking (Avg Rank) on classes of the open-source datasets in the LOVM Benchmark. For the third paradigm, we compare our method with ModelGPT [73].

**Evaluations.** We use Top-5 Recall and Kendall's Rank Correlation to measure the similarity between the predicted and the ground truth model rankings to evaluate the LOVM method's performance. We also calculate the sum of these two metrics to consider the method's comprehensive capability.

**Implementation Details.** For a fair comparison, SWAB follow ModelGPT [73] to sequentially extract a target dataset from each of the 23 datasets in the LOVM Benchmark and treat the remaining datasets as open-source datasets. Besides, SWAB adopts ModelGPT's approach of adding Gaussian noise to corrupt the target dataset's generated text embeddings. Since LOVM does not provide the specific image data for the 23 datasets, we download these datasets ourselves and adopt their standard data splits. Using the templates provided by LOVM, we construct classifiers and recalculate each VLM's zero-shot image classification accuracy on these datasets. Additionally, we utilize the code provided by LOVM to generate class-related text data using ChatGPT. For H-Score, NCE, LEEP, LogME, INB, and Avg Rank, we follow the practices of previous work and do not add noise to the model's inputs. To ensure reliable results, we conduct ten repeated experiments using random seeds from 1 to 10 and report the mean value and standard deviation of ModelGPT's performance and SWAB's performance in Table 1.

**Results Analysis.** From Table 1, we can draw the following conclusions: (1) Metric-based non-learning model selection methods such as LogME show poor performance on the LOVM Benchmark. This is primarily because such algorithms rely on the target dataset's images, thus the modality gap has a greater negative impact on them when using generated text data as a substitute for images. (2) Using open-source datasets is helpful for LOVM. We find that using open-source datasets in a non-learning way (*e.g.* INB, Avg Rank) or a learning way (*e.g.* ModelGPT) all helps LOVM, since their performance significantly surpasses that of methods not utilizing open-source datasets (*e.g.* LogME). (3) Despite leveraging more open-source datasets, the performance of Average Rank is worse than INB. This confirms our analysis of the Capability Gap, which suggests a discrepancy between the average ranking of a VLM and its ranking on a specific dataset. (4) Our SWAB *achieves the best performance across all evaluation metrics*. Notably, our final performance of $R_5 + \tau$ (0.822) represents a significant improvement of 14.8% over the original SoTA method ModelGPT (0.716).

## 4.2 Ablation Study

We conduct an ablation study to demonstrate that bridging the Modality Gap and Capability Gap are both essential for SWAB. Table 2 presents our experiment results, from which we can observe that SWAB achieves the best performance across all metrics when both gaps are bridged simultaneously. The ablation study confirms our analysis.

Table 2: **Ablation Study of** SWAB**.** SWAB-C, SWAB-M, and SWAB indicates only bridging the Capability Gap, only bridging the Modality Gap, and bridging both gaps in SWAB.

| Method | $R_5(\uparrow)$ | $\tau(\uparrow)$ | $R_5 + \tau(\uparrow)$ |
|---|---|---|---|
| SWAB-C | $0.487_{\pm 0.012}$ | $0.296_{\pm 0.018}$ | $0.783_{\pm 0.019}$ |
| SWAB-M | $0.474_{\pm 0.006}$ | $0.316_{\pm 0.019}$ | $0.790_{\pm 0.017}$ |
| SWAB | $\mathbf{0.504}_{\pm 0.000}$ | $\mathbf{0.318}_{\pm 0.002}$ | $\mathbf{0.822}_{\pm 0.002}$ |

### 4.3 Influence of Key Components in SWAB

**Will Bridge the Capability Gap Be Beneficial for VLM Selection?** We compare the LOVM performance directly using the VLM's average ranking on each class of open-source datasets and weighted-sum ranking based on transport matrix $\gamma^*$. The results are shown in Table 3. We can find that *utilizing class relevance to bridge the Capability Gap is beneficial for VLM's Model Selection.*

**Will Bridge the Modality Gap Be Beneficial for VLM Selection?** To eliminate the interference of other factors, we solely utilize the learning-based predicted rankings $\hat{r}_m^{\mathcal{T},(1)}$ in SWAB, and the input to the ranker model $f_m$ only consists of metrics calculated on the generated text data $\tilde{D}_{\mathcal{T}}$, which serves as substitutes for images. In this way, the method's performance depends solely on the quality of the generated text data $\tilde{D}_{\mathcal{T}}$. From the Table 4, we can find that the generated text data $\tilde{D}_{\mathcal{T}}$ become better substitutes for image data after bridging the Modality Gap.

Table 3: Results of $\hat{r}_m^{\mathcal{T},(2)}$ on the LOVM before and after bridging the Capability Gap.

| Method | $R_5(\uparrow)$ | $\tau(\uparrow)$ | $R_5 + \tau(\uparrow)$ |
|---|---|---|---|
| Average Rank | 0.443 | **0.246** | 0.689 |
| OT Weighted Rank | **0.513** | 0.217 | **0.730** |

Table 4: Results of $\hat{r}_m^{\mathcal{T},(2)}$ before and after bridging the Modality Gap (MG).

| Method | $R_5(\uparrow)$ | $\tau(\uparrow)$ | $R_5 + \tau(\uparrow)$ |
|---|---|---|---|
| Before Bridging MG | 0.216 | 0.061 | 0.277 |
| After Bridging MG | **0.371** | **0.080** | **0.451** |

**Which Kind of Gap Vectors Should We Use?** When utilizing the gap vectors from open-source datasets, we have two options: (1) Use the dataset-level mean gap vector calculated on the whole dataset's image-text pairs. (2) Use the class-level mean gap vector calculated on the corresponding class's image-text pairs. We hope that the gap vectors are as close to each other as possible so that their mean vector can substitute well for the whole set. Based on this idea, we calculate the statistics of the gap vectors within a dataset and within each class. We calculate three metrics which include: (1) the standard deviation of these gap vectors' magnitude; (2) the mean cosine similarity between these gap vectors and their corresponding mean gap vectors; and (3) the standard deviation of these cosine similarities. These metrics reflect the consistency of the gap vectors. Table 5 shows the results.

Table 5: Results of metrics measuring gap vectors' consistency belonging to the same dataset or the same class. M: Magnitude, D: Direction.

| Gap Vector | ImageNet | | |
|---|---|---|---|
| | M-Std($\downarrow$) | D-Mean($\uparrow$) | D-Std($\downarrow$) |
| Dataset Mean | 0.04 | 0.68 | 0.07 |
| Class Mean | **0.03** | **0.85** | **0.04** |

Table 6: Results of SWAB-M on the LOVM Benchmark using the dataset-level mean gap vectors and class-level mean gap vectors.

| Gap Vector | $R_5(\uparrow)$ | $\tau(\uparrow)$ | $R_5 + \tau(\uparrow)$ |
|---|---|---|---|
| Dataset Mean | 0.443 | 0.304 | 0.747 |
| Class Mean | **0.474** | **0.316** | **0.790** |

From the Table 5, we can find that the class-level gap vectors tend to be more consistent, which inspires us to use the class-level mean gap vectors. We also compare the results of SWAB-M on the LOVM Benchmark using the dataset-level mean gap vectors and the class-level mean gap vectors, respectively. The implementation details are the same as Table 2. Table 6 shows the results, which verifies that using the class-level mean gap vectors is a better choice.

## 5 Conclusion

We analyze and address two key challenges in Language-Only VLM Selection (LOVM), which are VLM's modality gap across different modal features and VLM's Capability gap between its overall and dataset-specific rankings. Our key insight is that we can reuse the model's useful statistics on open-source datasets to help the model selection on the target dataset. SWAB utilizes a transport matrix between classes of the target dataset and open-source datasets to transfer VLM's class-specific modality gap vectors and class-specific rank from open-source datasets to the target dataset, which mitigates the negative impacts of these two gaps. Experiment results on the LOVM benchmark show the superiority of our method.

## Acknowledgements

This work is partially supported by National Key R&D Program of China (2022ZD0114805), NSFC (62376118, 62250069, 62006112, 61921006), Collaborative Innovation Center of Novel Software Technology and Industrialization, CCF-Tencent Rhino-Bird Open Research Fund (RAGR20240101).

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

In the Appendix, we introduce more details about the LOVM Benchmark as well as our extensions to it. Besides, we introduce ModelGPT's implementation and our SWAB's implementation. We also provide more experimental results of SWAB. The structure of the Appendix is as follows:

- In section A, we introduce the relevant information of the 43 models and 23 datasets used in our experiments. We also introduce the evaluation metrics used in LOVM Benchmark [73].
- In section B, we introduce the metrics used in Equation 3.
- In section C, we introduce the related work of the paper.
- In section D, we provide some details on SWAB's implementation.
- In section E, we provide more experimental results of SWAB.

## A  LOVM Benchmark Details

LOVM Benchmark [73] consists of 35 pre-trained VLMs and 23 datasets, with a total of $35 \times 23 = 805$ evaluations. To further enhance the diversity of the VLM Zoo, we add some representative VLMs such as BLIP [28] and BEiT-3 [59] to expand the VLM Zoo. The final VLM Zoo contains a total of 43 models, with a total of $43 \times 23 = 989$ evaluations. For each evaluation, LOVM provides the VLM's zero-shot image classification accuracy on the corresponding dataset. Therefore, we can get the ground truth performance ranking of 43 VLMs on the 23 datasets.

### A.1  VLMs of LOVM Benchmark

To cover as many types of models as possible, the LOVM Benchmark uses OpenCLIP library [21] to get diverse VLMs. These VLMs differ from each other in terms of the model architecture (ResNet [17], Transformer [57], ConvNext [34]), the pre-trained dataset (OpenAI's Data [46], LAION 2b [47]), the training method (loss function/hyperparameter/data augmentation) and so on. Table 7 displays the relevant information of each VLM. We further add BLIP and BEiT-3 to the original VLM Zoo. It should be noted that in addition to the image-text pair data listed in the Table 7, BEiT-3's pre-training data also includes unimodal image dataset (ImageNet-21K) and text datasets (English Wikipedia, BookCorpus, OpenWebText, CC-News, Stories). The diversity of these VLMs ensures that the experimental results calculated on them can reflect the performance of the VLM model selection algorithm in real-world situations.

### A.2  Datasets of LOVM Benchmark

To cover as wide a distribution of image classification tasks as possible, the LOVM Benchmark collects 23 diverse datasets. These datasets differ from each other in terms of the number of categories, category semantics, image domains, and so on. Table 8 displays the relevant information of each dataset. The diversity of these tasks ensures that the experimental results calculated on them can reflect the performance of the VLM model selection method in real-world situations.

### A.3  Evaluation Metrics of LOVM Benchmark

In LOVM, our aim is to maximize the rank similarity between the prediction of VLMs' ranking $\hat{r}_\mathcal{T} = \{\hat{r}_m^\mathcal{T}\}_{m=1}^M$ and VLMs' ground truth ranking $r_\mathcal{T} = \{r_m^\mathcal{T}\}_{m=1}^M$ on the target dataset, especially the rank similarity of the top 5 VLMs in $\hat{r}_\mathcal{T}$ and $r_\mathcal{T}$. This is because we tend to focus only on whether those appropriate models can be chosen. To ensure fair comparability, we follow the evaluation metrics used by LOVM [73] to assess different model selection methods and directly utilize the code provided by LOVM [73] to calculate these metrics:

- **Top-5 Recall ($R_5$)** – Top-5 Recall $R_5$ measures the model selection algorithm's accuracy in identifying the true top five best-performing models within its predicted top five models. The calculation method is shown in Equation 11. Here $\text{IND}(\hat{r}_\mathcal{T}^5)$ and $\text{IND}(r_\mathcal{T}^5)$ indicates the model indices sets of the top 5 VLMs in $\hat{r}_\mathcal{T}$ and $r_\mathcal{T}$, respectively. A Top 5 Recall closer to 1 signifies greater accuracy in the predicted rankings.

$$F = \text{IND}(\hat{r}_\mathcal{T}^5) \cap \text{IND}(r_\mathcal{T}^5). \tag{10}$$

$$R_5 = \frac{|F|}{5}. \tag{11}$$

Table 7: The detailed information of 43 models used in the LOVM Benchmark. Some of the information in the table comes from [73].

| ID | Model | Name | Dataset | Name |
|----|-------|------|---------|------|
| 1 | RN50 | RN50 | openai | WIT |
| 2 | RN101 | RN101 | openai | WIT |
| 3 | RN50x4 | RN50x4 | openai | WIT |
| 4 | RN50-16 | RN50x16 | openai | WIT |
| 5 | RN50x64 | RN50x64 | openai | WIT |
| 6 | ViT-B-32 | ViT-B/32 | laion400m_e31 | L400m |
| 7 | ViT-B-32 | ViT-B/32 | laion400m_e32 | L400m |
| 8 | ViT-B-32-quickgelu | ViT-B/32 | laion400m_e32 | L400m |
| 9 | ViT-B-32 | ViT-B/32 | openai | WIT |
| 10 | ViT-B-32 | ViT-B/32 | laion2b_s34b_b79k | L2b-b |
| 11 | ViT-B-32 | ViT-B/32 | laion2b_e16 | L2b-c |
| 12 | ViT-B-16 | ViT-B/16 | laion400m_e32 | L400m |
| 13 | ViT-B-16 | ViT-B/16 | openai | WIT |
| 14 | ViT-B-16-240 | ViT-B/16-240 | laion400m_e32 | L400m |
| 15 | ViT-L-14 | ViT-L/14 | laion400m_e31 | L400m |
| 16 | ViT-L-14 | ViT-L/14 | laion400m_e32 | L400m |
| 17 | ViT-L-14 | ViT-L/14 | laion2b_s32b_b82k | L2b-b |
| 18 | ViT-L-14 | ViT-L/14 | openai | WIT |
| 19 | ViT-L-14-336 | ViT-L/14-336 | openai | WIT |
| 20 | ViT-G-14 | ViT-G/14 | laion2b_s12b_b42k | L2b-a |
| 21 | ViT-G-14 | ViT-G/14 | laion2b_s34b_b88k | L2b-a |
| 22 | ViT-H-14 | ViT-H/14 | laion2b_s32b_b79k | L2b-b |
| 23 | coca_ViT-B-32 | CoCa-ViT-B/32 | laion2b_s13b_b90k | L2b-c |
| 24 | coca_ViT-B-32 | CoCa-ViT-B/32 | mscoco_finetuned_laion2b_s13b_b90k | L2b-c + coco |
| 25 | coca_ViT-L-14 | CoCa-ViT-L/14 | laion2b_s13b_b90k | L2b-c |
| 26 | coca_ViT-L-14 | CoCa-ViT-L/14 | mscoco_finetuned_laion2b_s13b_b90k | L2b-c + coco |
| 27 | convnext_base | ConvNEXT-B | laion400m_s13b_b51k | L400m-c |
| 28 | convnext_base_w | ConvNEXT-BW | laion2b_s13b_b82k | L2b-d |
| 29 | convnext_base_w | ConvNEXT-BW | laion2b_s13b_b82k_augreg | L2b-e |
| 30 | convnext_base_w | ConvNEXT-BW | laion_aesthetic_s13b_b82k | L2b-f |
| 31 | convnext_base_w_320 | ConvNEXT-BW-320 | laion_aesthetic_s13b_b82k | L2b-f |
| 32 | convnext_base_w_320 | ConvNEXT-BW-320 | laion_aesthetic_s13b_b82k_augreg | L2b-g |
| 33 | convnext_large_d | ConvNEXT-LD | laion2b_s26b_b102k_augreg | L2b-h |
| 34 | convnext_large_d_320 | ConvNEXT-LD-320 | laion2b_s29b_b131k_ft | L2b-i |
| 35 | convnext_large_d_320 | ConvNEXT-LD-320 | laion2b_s29b_b131k_ft_soup | L2b-j |
| 36 | BLIP_retrieval_base_coco | BLIP_retrieval_base_coco | COCO+VG+CC+CC12M+SBU | BLIP-Dataset |
| 37 | BLIP_retrieval_base_f30k | BLIP_retrieval_base_f30k | COCO+VG+CC+CC12M+SBU+Flickr30k | BLIP-Dataset-f |
| 38 | BLIP_retrieval_large_coco | BLIP_retrieval_large_coco | COCO+VG+CC+CC12M+SBU | BLIP-Dataset |
| 39 | BLIP_retrieval_large_f30k | BLIP_retrieval_large_f30k | COCO+VG+CC+CC12M+SBU+Flickr30k | BLIP-Dataset-f |
| 40 | BEiT-3_retrieval_base_coco | BEiT-3_retrieval_base_coco | CC12M+CC3M+SBU+COCO+VG | BEiT-Dataset |
| 41 | BEiT-3_retrieval_base_f30k | BEiT-3_retrieval_base_f30k | CC12M+CC3M+SBU+COCO+VG+Flickr30k | BEiT-Dataset-f |
| 42 | BEiT-3_retrieval_large_coco | BEiT-3_retrieval_large_coco | CC12M+CC3M+SBU+COCO+VG | BEiT-Dataset |
| 43 | BEiT-3_retrieval_large_f30k | BEiT-3_retrieval_large_f30k | CC12M+CC3M+SBU+COCO+VG+Flickr30k | BEiT-Dataset-f |

Table 8: Detailed information of 23 tasks used in the LOVM Benchmark. This table comes from [73].

| Dataset | Classes | Task | Domain |
|---------|---------|------|--------|
| Imagenet [8] | 1000 | classification | natural image |
| SUN397 [60] | 397 | scene und. | natural image |
| Country211 [46] | 211 | geolocation | natural image |
| Stanford Cars [25] | 196 | classification | natural image |
| Flowers102 [42] | 102 | classification | natural image |
| CIFAR100 [26] | 100 | classification | natural image |
| DTD [5] | 46 | classification | textural image |
| RESISC45 [4] | 45 | classification | satellite images |
| GTSRB [50] | 43 | classification | natural image |
| Oxford Pets [44] | 37 | classification | natural image |
| VOC2007 [10] | 20 | classification | natural image |
| STL10 [6] | 10 | classification | natural image |
| EuroSAT [19] | 10 | classification | satellite images |
| MNIST [27] | 10 | classification | hand-writing |
| SVHN [41] | 10 | OCR | natural image |
| CLEVR-C [23] | 8 | object counting | natural image |
| CLEVR-D [23] | 8 | distance est. | natural image |
| FER2013 [16] | 7 | fac. exp. rec. | natural image |
| DMLab [69] | 6 | distance est. | synthetic |
| Retinopathy [24] | 5 | classification | retina scan |
| KITTI [15] | 4 | distance est. | natural image |
| PCam [58] | 2 | classification | histopathology |
| Rendered SST2 [46] | 2 | OCR | text image |

- **Kendall's Rank Correlation ($\tau$)** – Kendall's Rank Correlation $\tau$ measures the ranking consistency between two ranking lists. We follow LOVM [73] to focus on the VLMs within the intersection of the top 5 VLMs in $\hat{r}_{\mathcal{T}}$ and $r_{\mathcal{T}}$ and use $\tau$ to evaluate a model selection method's capability.

# B  ModelGPT Details

ModelGPT is a method proposed by LOVM [73]. In this section, we introduce the metrics that ModelGPT used in Equation 3.

## B.1  The Generation Process of Auxiliary Text Samples

ModelGPT [73] utilizes ChatGPT [43] to generate auxiliary text data by designing prompts to query ChatGPT. This extra text data mainly includes the Captions Dataset and the Synonyms Dataset.

**Captions Dataset.** ModelGPT uses the following prompt to guide LLM to generate realistic and confusing text data corresponding to the user-provided classes. The reason for requiring ChatGPT to generate confusing texts is to increase the classification difficulty of the text data, thereby enhancing its ability to distinguish the performance of different models.

> Generate long and confusing image captions for the {domain} domain, which
> will be used to evaluate a Vision-Language Model's {task} performance.
> Generate 50 captions for {classname}:

We show some generated auxiliary text examples. For example, in the category of dog, one of the text samples generated by ChatGPT is "An adorable dog perfect for cuddles and playtime." ModelGPT collects the results from this prompt to form the captions dataset, $\boldsymbol{D}^{\mathrm{cap}}$.

**Synonyms Dataset.** ModelGPT uses synonyms to evaluate VLM's text encoder. For example, we expect an excellent VLM to extract similar embeddings for the words "chair" and "seat". The prompt to guide LLM to generate synonyms is as follows.

> Please list the superclasses/synonyms for {classname}. For example:
> chair: [furniture, seat, bench, armchair, sofa]
> {classname}:

ModelGPT collects the results from this prompt to form the synonyms dataset, $\boldsymbol{D}^{\mathrm{syn}}$.

## B.2  Text-Derived Scores

ModelGPT uses six metrics for model selection, which can be divided into **Text Classification scores** and **Dataset Granularity scores**. **Text Classification scores** include the *Text top-1 accuracy score* and *Text f1-score*. While **Granularity scores** include the *Fisher criterion*, *Silhouette score*, *Class Dispersion score* and *Synonym Consistency score*. Here we focus on introducing the various metrics included in the **Granularity scores**. We refer to the relevant content in LOVM.

**Fisher Criterion** $\phi_{\mathrm{fisher}}$. The Fisher score measures the closeness of VLM's text classifier to one another. Equation 12 shows the calculation process of it where $\hat{\boldsymbol{t}}_i$ is the text classifier of the $i$-th class derived using the prompt ensemble strategies proposed in [46], $\theta(\cdot, \cdot)$ is a function that calculates the cosine similarity between two vectors, and $|C|$ is the number of classes.

$$\phi_{\mathrm{fisher}} = \frac{1}{|C|} \sum_{j=1}^{|C|} \max_{i, i \neq j} \left[ \theta(\hat{\boldsymbol{t}}_i, \hat{\boldsymbol{t}}_j) \right] . \tag{12}$$

**Silhouette Score** $\varphi_{\mathrm{sil}}$. The Silhouette Score measures the separation of different-class samples in the caption dataset $\boldsymbol{D}^{\mathrm{cap}}$. To calculate it, ModelGPT averages the cosine similarity of captions to the nearest other class's classifier by:

$$\varphi_{\mathrm{sil}} = \frac{1}{|C|} \sum_{j=1}^{|C|} \max_{i, i \neq j} \left[ \frac{1}{N} \sum_{k=1}^{N} \theta(\boldsymbol{D}^{\mathrm{cap}}[j]_k, \hat{\boldsymbol{t}}_i) \right] . \tag{13}$$

where $\hat{\boldsymbol{t}}_i$ is the text classifier of the $i$-th class derived using the prompt ensemble strategies proposed in [46], $\theta(\cdot, \cdot)$ is a function that calculates the cosine similarity between two vectors, and $|C|$ is the number of classes. $\boldsymbol{D}^{\mathrm{cap}}[j]_k$ representing sample $k$ of class $j$ in the caption dataset $\boldsymbol{D}^{\mathrm{cap}}$. There is a total of $N$ such samples for each class.

**Class Dispersion Score** $\rho_{\mathrm{disp}}$. Class Dispersion Score quantifies the degree of same-class tightness or data cone radius, which is calculated using the following Equation:

$$\rho_{\mathrm{disp}} = \frac{1}{|C|N} \sum_{i=1}^{|C|} \sum_{k=1}^{N} \theta(\boldsymbol{D}^{\mathrm{cap}}[i]_k, \hat{\boldsymbol{t}}_i). \tag{14}$$

The definitions of all symbols in Equation 14 are consistent with those in Equation 13.

**Synonym Consistency Score** $\gamma_{\mathrm{syn}}$. Synonym consistency allows us to evaluate the degree of content shift between the VLMs' pre-training and target dataset. The calculation process is shown as follows:

$$\gamma_{\mathrm{syn}} = \frac{1}{|C|N} \sum_{i=1}^{|C|} \sum_{k=1}^{N} \theta(\boldsymbol{D}^{\mathrm{syn}}[i]_k, \hat{\boldsymbol{t}}_i). \tag{15}$$

The definitions of $\hat{\boldsymbol{t}}_i$, $\theta(\cdot, \cdot)$, $|C|$ and $N$ in Equation 15 are consistent with those in Equation 13. $\boldsymbol{D}^{\mathrm{syn}}[i]_k$ representing sample $k$ of class $i$ in the synonym dataset $\boldsymbol{D}^{\mathrm{syn}}$.

## C Related Work

**Vision-Language Models.** Deep learning has achieved competitive performance [39, 74, 30] and the vision-language model is a new research hotspot of it. Vision-Language Models represent a class of multimodal models adept at correlating textual and visual information. These VLMs can comprehend rich semantics and possess strong generalization capabilities. Hence, they are often used in data-limited scenarios [61, 29]. Many VLMs are pre-trained or fine-tuned on extensive text-image pairs using loss functions such as contrastive loss, endowing them with powerful text-image matching capability. Prominent VLMs include CLIP [46], ALIGN [22], FLAVA [48], Florence [68], and CoCa [67]. These VLMs possess robust zero-shot image classification capabilities [46], which enables its widespread application in tasks characterized by long-tail distributions or those where collecting substantial training data is challenging, such as medical image analysis. Some work [40, 64] has also shown that by incorporating external knowledge, the zero-shot capabilities of CLIP can be further enhanced. In recent years, the number of open-source VLMs has been increasing [21]. Previous work [73] has pointed out that different VLMs possess varying image classification capabilities. This indicates that the performance of VLMs can vary significantly across different tasks and domains. These models with diverse capabilities constitute a VLM Zoo rich in knowledge. This VLM Zoo enables us to utilize different VLMs for various classification tasks, thereby changing the paradigm of using a single VLM to complete diverse classification tasks. This paper focuses on selecting the most suitable VLM for the target task from the VLM Zoo.

**Model Selection.** In recent years, how to select the most suitable model for the target task from the model zoo has received widespread attention. For example, a series of works on learnware [53, 52, 71, 72, 33, 51, 54] have attempted to solve this problem. The essence of the model selection problem lies in measuring the transferability of the model to the target task. Previous model selection methods [55, 56, 65, 66, 9, 20] evaluate the PTM's transferability by performing a forward pass of the PTM on the target task's data and calculating the metric of transferability based on the forward pass's result. For example, H-Score [2], NCE [55], LEEP [56], LogME [65] estimate transferred log-likelihood, negative conditional entropy, log expectation, marginalized likelihood to obtain proxy metric of transferability, respectively. Some new methods, such as Task2Vec [1] and Model Spider [70], generate representation vectors for both the models and the tasks and measure the transferability of the model to the task by calculating the similarity between these vectors. However, VLMs are typically used in zero-shot or few-shot scenarios, where we are unable to obtain a large amount of data for the target task in advance, making traditional model selection approaches unsuitable for VLMs. Additionally, previous methods have mainly focused on single-modal models, overlooking the characteristics of VLMs. Therefore, in this paper, we concentrate on designing model selection algorithms that are suitable for scenarios with limited data and take into account the characteristics of VLMs.

# D  Implementation Details of SWAB

In this section, we provide some details on the implementation of SWAB, which are not mentioned in the main text due to space constraints.

## D.1  Filtering the Open-Source Tasks' Classes

The statistics of classes unrelated to the target class are generally not valuable for reuse. Meanwhile, when the number of classes $|C_{\mathcal{S}}|$ in open-source datasets $\mathcal{S}$ is large, solving the optimal transport problem in Equation 5 can be time-consuming (as current optimal transport toolkits generally compute via CPU). To reduce the runtime of optimal transport, we can first filter the classes $C_{\mathcal{S}}$. Consider that only statistics of classes relevant to the target dataset are helpful. Therefore, we can filter out the classes in $C_{\mathcal{S}}$ that are irrelevant to the target dataset $\mathcal{T}$ based on the class-level textual semantic similarity between the open-source datasets' classes and the target dataset's classes. This process is shown in the following Equation:

$$\boldsymbol{S}_{ij} = \frac{\phi(c_i^{\mathcal{S}})^{\top}\phi(c_j^{\mathcal{T}})}{\|\phi(c_i^{\mathcal{S}})\| \cdot \|\phi(c_j^{\mathcal{T}})\|} \ . \tag{16}$$

$$C_{\mathcal{S}}^{'} = \{c_i^{\mathcal{S}} | \max(\boldsymbol{S}_{i,:}) > \lambda\}, \ |C_s^{'}| = k_{\mathcal{S}}^{'}. \tag{17}$$

Here $\boldsymbol{S}_{i,:}$ refers to the $i$-th row of the semantic similarity matrix calculated using Equation 16, which represents the vector formed by the similarity between the $i$-th class $c_i^{\mathcal{S}}$ in $C_{\mathcal{S}}$ and each class $C_{\mathcal{T}} = \{c_1^{\mathcal{T}}, \cdots, c_{k_{\mathcal{T}}}^{\mathcal{T}}\}$ of the target task. $\lambda$ is a threshold and we set $\lambda = 0.5$. $k_{\mathcal{S}}^{'}$ refers to the number of classes in the filtered set $C_{\mathcal{S}}^{'}$. Then we use the filter classes $C_{\mathcal{S}}^{'}$ to calculate the transport matrix $\gamma^* \in \mathbb{R}^{k_{\mathcal{S}}^{'} \times k_{\mathcal{T}}}$ and continue with the following steps.

## D.2  Using Partial Optimal Transport for Bridging the Capability Gap

Partial optimal transport extends the optimal transport framework, enabling the selective transfer of elements from a source to a target distribution, rather than moving all elements. Its optimization problem is defined as in Equation 18. Here $mass$ refers to the total amount of mass actually be transferred. We set $mass = 0.9$ in our implementation.

$$\begin{aligned} \boldsymbol{\gamma}^* = \ &\underset{\boldsymbol{\gamma} \in \mathbb{R}_+^{k_{\mathcal{S}}^{'} \times k_{\mathcal{T}}}}{\operatorname{argmin}} \sum_{i,j} \gamma_{i,j} \operatorname{cost}_{i,j} \\ &\text{s.t. } \boldsymbol{\gamma}\mathbf{1} \leq \boldsymbol{u}; \ \boldsymbol{\gamma}^T\mathbf{1} \leq \boldsymbol{v}; \ \gamma_{i,j} \geq 0; \\ &\mathbf{1}^T\boldsymbol{\gamma}^T\mathbf{1} = mass \leq \min\{\|\boldsymbol{u}\|_1, \|\boldsymbol{v}\|_1\} \ . \end{aligned} \tag{18}$$

We found that when using Equation 8 to bridge the Capability Gap, the transport matrix $\boldsymbol{\gamma}^*$ obtained using partial optimal transport yields better results than the one obtained using the original optimal transport via solving the Equation 5. Therefore, in our implementation, we use the transport matrix derived from partial optimal transport to bridge the capability gap. This also indicates that when estimating VLM's statistics on the target dataset, different types of statistics have different preferences for the estimation methods used. This variability is worth further investigation.

## D.3  Data Normalization in Bridging the Modality Gap.

When bridging the Modality Gap as described in Section 3.2, we find that applying z-score normalization to the text and image features used in this process yields better results. Therefore, in our implementation, we normalize the features of all text and image samples during the modality bridging process using the following Equation:

$$\boldsymbol{z} = \frac{\boldsymbol{x} - \boldsymbol{\mu}}{\boldsymbol{\sigma}}. \tag{19}$$

Here $\boldsymbol{x} \in \mathbb{R}^d$ represents the image sample's or text sample's feature, while $\boldsymbol{\mu} \in \mathbb{R}^d$ and $\boldsymbol{\sigma} \in \mathbb{R}^d$ are calculated using the features of all samples of the same modality within its respective dataset.

**Algorithm 1** SWAB

---

1: **Input:** Target dataset's class names $C_{\mathcal{T}}$, open-source datasets' class names $C_{\mathcal{S}}$, open-source datasets' images $D_{\mathcal{S}}^I$.
2: Use ChatGPT to generate auxiliary text data $\tilde{D}_{\mathcal{S}}$ and $\tilde{D}_{\mathcal{T}}$ based on $C_{\mathcal{S}}$ and $C_{\mathcal{T}}$.
3: Calculate VLM's class-level zero-shot image classification rankings $\{r_{m,i}^{\mathcal{S}}\}_{m=1,i=1}^{m=M,i=k_{\mathcal{S}}}$ and class-level gap vectors $\{\boldsymbol{g}_{m,i}^{\mathcal{S}}\}_{m=1,i=1}^{m=M,i=k_{\mathcal{S}}}$.
4: **for** $k_{\mathcal{S}}$ datasets **do**
5:    Calculate textual similarity between the current dataset's class names and other open-source datasets' class names to construct a cost matrix.
6:    Solve Optimal Transport Problem based on the cost matrix to get transport matrix $\boldsymbol{\gamma}^*$.
7:    Use $\boldsymbol{\gamma}^*$ and other open-source datasets' class-level gap vectors to predict the current dataset's class-level gap vectors. Add the predicted class-level gap vectors to the corresponding text data's feature of $\tilde{D}_{\mathcal{S}}$ to get modified text data.
8: **end for**
9: Calculate textual similarity between open-source datasets' class names $C_{\mathcal{S}}$ and target dataset's class names $C_{\mathcal{T}}$ to construct cost matrix.
10: Solve Optimal Transport Problem based on the cost matrix to get transport matrix $\boldsymbol{\gamma}^*$.
11: Use $\boldsymbol{\gamma}^*$ and $\{\boldsymbol{g}_{m,i}^{\mathcal{S}}\}_{m=1,i=1}^{m=M,i=k_{\mathcal{S}}}$ to predict the class-level vectors $\{\hat{\boldsymbol{g}}_{m,i}^{\mathcal{T}}\}_{m=1,i=1}^{m=M,i=k_{\mathcal{T}}}$ of the target dataset. Add $\{\hat{\boldsymbol{g}}_{m,i}^{\mathcal{T}}\}_{m=1,i=1}^{m=M,i=k_{\mathcal{T}}}$ to corresponding text data's feature of $\tilde{D}_{\mathcal{T}}$ to get modified text data.
12: Use open-source datasets' modified text data and VLMs' ground truth zero-shot image classification performance to train the ranker model $f_m$.
13: Use the ranker model $f_m$ to predict VLMs' rankings $\{\hat{r}_m^{\mathcal{T},(1)}\}_{m=1}^{m=M}$ on the target dataset based on the target dataset's modified text data.
14: Use $\boldsymbol{\gamma}^*$ and $\{\boldsymbol{r}_{m,i}^{\mathcal{S}}\}_{m=1,i=1}^{m=M,i=k_{\mathcal{S}}}$ to predict the VLMs' class-level zero-shot image classification rankings $\{\hat{\boldsymbol{r}}_{m,i}^{\mathcal{T}}\}_{m=1,i=1}^{m=M,i=k_{\mathcal{T}}}$ of the target dataset.
15: Calculate the average prediction rankings across all classes in the target dataset $\{\hat{r}_m^{\mathcal{T},(2)}\}_{m=1}^{m=M}$ of $\{\hat{\boldsymbol{r}}_{m,i}^{\mathcal{T}}\}_{m=1,i=1}^{m=M,i=k_{\mathcal{T}}}$ for each VLM as the VLM's overall prediction ranking on the target dataset.
16: Ensemble $\{\hat{r}_m^{\mathcal{T},(1)}\}_{m=1}^{m=M}$ and $\{\hat{r}_m^{\mathcal{T},(2)}\}_{m=1}^{m=M}$ to get final predicted rankings $\{\hat{r}_m^{\mathcal{T},ens}\}_{m=1}^{m=M}$.

---

### D.4  Pseudo Code of SWAB

Algorithm 1 shows the pseudo-code of SWAB.

## E  More Experiment Results

In this section, we provide more experimental results of SWAB.

### E.1  Bridging the Modality Gap Leads to Better Image Proxies

In Section 2.3 and Figure 2, we analyze whether generated text data can act as good image proxies. Our conclusion is that due to the Modality Gap, text samples cannot directly serve as an effective substitute for images in model evaluation. To demonstrate that our method SWAB can bridge this Modality Gap and thereby make text samples a better substitute for images, we conduct the following experiment.

From the Figure 4, it is evident that the predicted model accuracy calculated using the modified text samples is closer to the true model accuracy compared to that calculated with the original text samples. This suggests that bridging the Modality Gap leads to better image proxies.

We use ImageNet as our dataset. First, we employ the method introduced in Section 3.2 to predict the gap vectors for each class of the target dataset based on gap vectors calculated on open-source datasets. Then, we add the corresponding class's predicted gap vectors to the generated text data of ImageNet to bridge the modality gap. Finally, we calculate the zero-shot classification accuracy of different models on these modified text data. To measure the consistency between the predicted Top-1

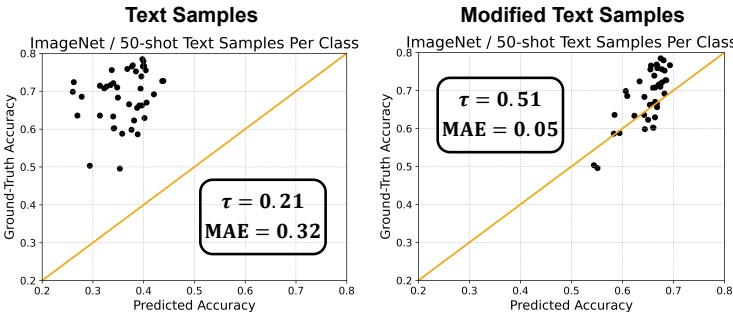

Figure 4: Comparison of the consistency metrics between the accuracy calculated using text data before and after bridging the gap and the model's true accuracy. After bridging the modality gap, the text data act as better substitutes for image data to evaluate the model's performance.

accuracy and the true image classification accuracy, we calculate the Kendall Rank Correlation ($\tau$, higher is better) and Mean Absolute Error (MAE, lower is better). We compare the consistency metrics of text data and modified text data. It can be observed that the consistency metrics of modified text data are better, which proves our method can reduce the gap between the generated text data and the image data.

## E.2 Analysis of the Modality Gap in BLIP and BEiT-3

We expand the VLM Zoo provided by LOVM by adding various variants of BLIP and BEiT-3 to enhance the diversity of the VLM Zoo. We find that the zero-shot image classification performance of the base models of BLIP and BEiT-3 is poor. Therefore, we use the retrieval version of the models provided officially. These models are obtained by fine-tuning the base models on COCO and Flickr30k through contrastive learning, thereby possessing better zero-shot image classification performance. During the pre-training of BLIP and BEiT-3, they may have used multiple loss functions in addition to the contrastive loss (BLIP), or perhaps they did not use contrastive loss at all (BEiT-3). Therefore, the existence of the Modality Gap in the feature spaces of these two models requires further investigation.

We use image samples from the STL-10 dataset and class-related text samples generated by ChatGPT. For each modality, we randomly extract 200 samples and calculate image-to-image, text-to-text, and image-to-text similarities, and plot histograms. We also observe the presence of Modality Gaps in different models through UMAP visualization [38]. Figure 5 and Figure 6 show the experiment result. Through our experiments, we verify that the feature spaces of the BLIP and BEiT-3 retrieval models still exhibit the Modality Gap phenomenon.

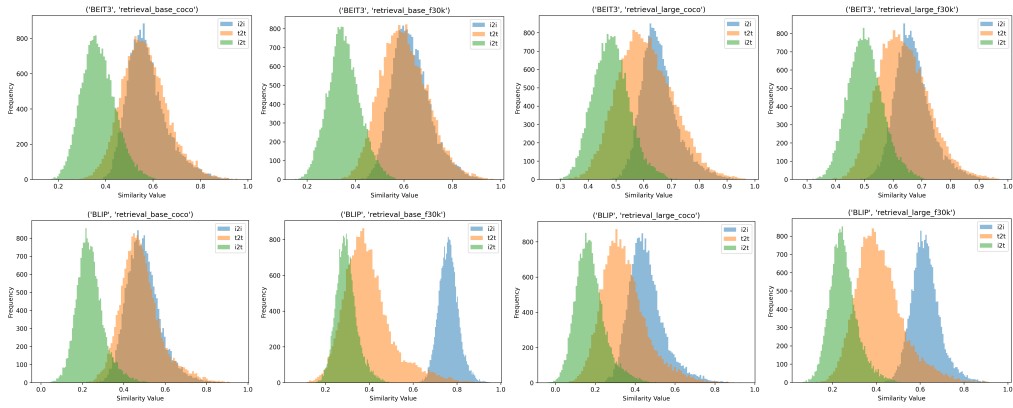

Figure 5: The distribution of image-to-image (i2i) cosine similarity, text-to-text (t2t) cosine similarity, and image-to-text (i2t) cosine similarity values for different BEiT-3 and BLIP models.

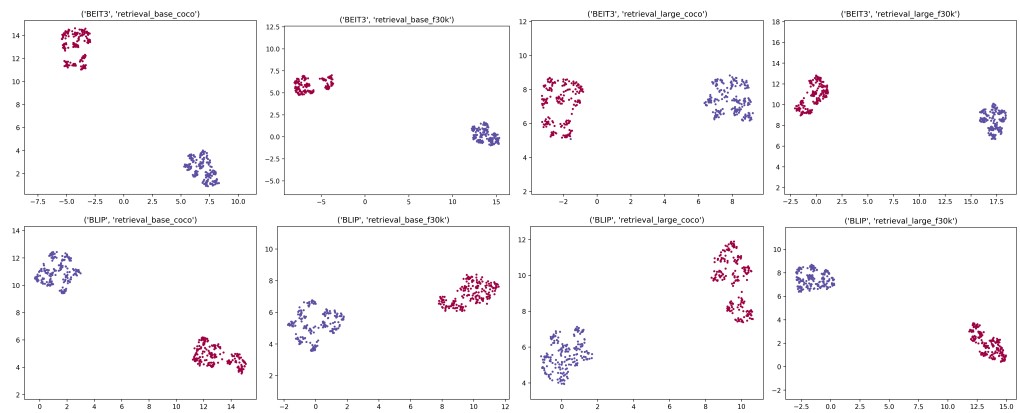

Figure 6: UMAP visualization of image sample features and text sample features from different BEiT-3 and BLIP models.

### E.3 Experiment Result on LOVM's original VLM Zoo

We have provided experimental results on the VLM Zoo originally provided by LOVM [73]. This VLM Zoo contains 35 VLMs. Table 9 shows the experimental results, and we can see that SWAB achieves the best performance across all evaluation metrics.

Table 9: **Results on LOVM's original VLM Zoo.** We evaluate our method across 23 datasets and 35 pre-trained VLMs. The results are averaged over all datasets.

| Methods | H-Score | NCE | LEEP | LogME | INB | Avg Rank | ModelGPT | SWAB |
|---|---|---|---|---|---|---|---|---|
| $R_5(\uparrow)$ | 0.183 | 0.235 | 0.139 | 0.200 | 0.443 | 0.443 | $0.446_{\pm 0.004}$ | $\mathbf{0.498}_{\pm 0.005}$ |
| $\tau(\uparrow)$ | 0.000 | 0.000 | 0.014 | -0.014 | 0.267 | 0.261 | $0.272_{\pm 0.010}$ | $\mathbf{0.310}_{\pm 0.012}$ |
| $R_5 + \tau(\uparrow)$ | 0.183 | 0.235 | 0.153 | 0.186 | 0.710 | 0.704 | $0.718_{\pm 0.013}$ | $\mathbf{0.808}_{\pm 0.011}$ |

### E.4 Per-Dataset Experiment Result

We present the per-dataset performance comparison between our methods SWAB and ModelGPT on various datasets of LOVM benchmark in Table 10.

Table 10: **LOVM Benchmark (top-1 accuracy).** We compare the per-dataset experiment result (fixing the random seed) between ModelGPT and SWAB.

| | | Stanford Cars | CIFAR100 | CLEVR-DIST. | CLEVR-COUNT | Country211 | Retinopathy | DMLab | DTD | EuroSAT | FER2013 | Flowers102 | GTSRB | ImageNet | KITTI | MNIST | PCam | Oxford Pets | Rendered SST2 | RESISC45 | STL10 | SUN397 | SVHN | VOC2007 | Mean |
|---|---|---|---|---|---|---|---|---|---|---|---|---|---|---|---|---|---|---|---|---|---|---|---|---|---|
| $R_5$ | ModelGPT | 0.80 | 0.80 | 0.00 | 0.40 | 0.60 | 0.00 | 0.00 | 0.80 | 0.60 | 0.40 | 0.80 | 0.60 | 0.80 | 0.00 | 0.00 | 0.00 | 0.80 | 0.20 | 0.80 | 0.20 | 0.60 | 0.60 | 0.60 | 0.452 |
| | SWAB | 0.80 | 0.80 | 0.00 | 0.60 | 0.40 | 0.20 | 0.00 | 0.80 | 0.60 | 0.60 | 0.80 | 0.60 | 1.00 | 0.00 | 0.40 | 0.00 | 0.80 | 0.20 | 1.00 | 0.20 | 0.60 | 0.60 | 0.60 | **0.504** |
| $\tau$ | ModelGPT | 0.67 | 0.67 | 0.00 | 0.00 | -0.33 | 0.00 | 0.00 | 0.67 | 1.00 | 0.00 | 0.67 | 1.00 | 0.67 | 0.00 | 0.00 | 0.00 | 0.00 | 0.00 | 0.67 | 0.00 | -1.00 | 0.33 | 1.00 | 0.261 |
| | SWAB | 0.67 | 0.91 | 0.00 | 0.33 | 0.00 | 0.00 | 0.00 | 0.55 | 1.00 | 0.33 | 0.33 | 1.00 | 0.89 | 0.00 | 0.00 | 0.00 | 0.33 | 0.00 | 0.67 | 0.00 | -0.82 | 0.82 | 0.33 | **0.320** |

