# OpenReview forum: "Bridge the Modality and Capability Gaps in Vision-Language Model Selection"
_NeurIPS.cc/2024/Conference — NeurIPS 2024 poster_

### Official Review · Reviewer_mpun · 2024-07-06

**Soundness:** 2
**Presentation:** 3
**Contribution:** 3
**Rating:** 5
**Confidence:** 4

**Summary:**

This paper considers a zero-shot image classification strategy by selecting the most appropriate Pre-Trained VLM from the VLM Zoo, relying solely on the text data of the target dataset without access to the dataset’s images. Two challenges, i.e., the “Modality Gap” across two different modalities, and the "Capability Gap" between the VLM’s overall ranking and its ranking for target dataset, hinder the appropriate selection of the different VLMs. To address the challenges, this paper adopts a transportation matrix to capture the relevance between open-source and target datasets, and also uses this matrix to transfer useful statistics of VLMs from open-source datasets to the target dataset. Extensive experiments on image classification datasets demonstrate the effectiveness of the proposed method.

**Strengths:**

Originality:
The main novelty of this paper is to utilize Language-Only VLM Selection to address the zero-shot image classification. This is the first time that the Language-Only VLM Selection is applied in the zero-shot scenario where the missing of image data makes it difficult to directly select the VLMs.
Quality:
This paper has well motivated methods and extensive experiments.
Clarity:
The paper is well written with clear description and readable figures.
Significance:
The two key issues of modality gap and capability gap are considered in this paper. In addition, optimal transport-based methods are proposed to alleviate the issues and the experimental results on multiple VLMs and image classification datasets show the effectiveness of them.

**Weaknesses:**

The main weakness is that the proposed VLMs selection strategy relies on Open-Source Datasets’ Data, which restricts the practicability of the proposed method. For the zero-shot image classification task, using the aggregated knowledge from multiple VLMs is ok, but using open-source datasets may violate the zero-shot data restrictions, or even lead to benchmark leakage. In addition, the impact of the open-source datasets is not analyzed. For example, if the categories in the open-source datasets are very different from the ones in the target zero-shot classification tasks, whether the performance of the proposed method is still remarkable?

**Questions:**

A comprehensive analysis on the impact of the open source datasets may improve the rationality of the proposed method for addressing the zero-shot image classification task.

**Limitations:**

This paper does not include the limitation discussion. Practically, at least the issues raised in the weakness section should be considered if they cannot be addressed appropriately in current version.

---

> ### Author Rebuttal · Authors · 2024-08-06
>
> We appreciate the reviewer's comments. Below are our responses:
>
> * Q1：Use open-source datasets may violate zero-shot data restrictions.
> * A1:
>
>     1. Firstly, we want to emphasize that our method focuses on how to use open-source datasets to help us select the appropriate models from the VLM Zoo for the current task, rather than using open-source datasets to enhance the zero-shot image classification capabilities of pre-trained VLMs. For the pre-trained VLMs we selected, we directly complete the image classification tasks on the target datasets without using any additional data or undergoing any extra training process.  **Therefore, we do not introduce additional data when the VLM performs image classification tasks, and there is no issue with data leakage.**
>
>     2. Secondly, the open-source datasets we use are all common and widely used. And since the pre-traning phase of VLMs have also used a lot of open-source data, it is likely that VLM's pre-training data has already included the data similar to the open-source datasets we use. Therefore, we believe that our use of common open-source datasets is reasonable.  It is important to emphasize that the previous SoTA method, ModelGPT[1], used the same open-source datasets as we did, ensuring that our comparison with their results is fair. Moreover, we believe that in the era of large models, a developmental trend is that we should make the most of the data and resources that are easily accessible to us.
>
> * Q2: Whether the performance of the proposed method is still remarkable if the categories in the open-source datasets are very different from the ones in the target zero-shot classification tasks.
> * A2:
>
>     The key insight of our method is to leverage the category relatedness to reuse the model's performance evaluation results from previous tasks. Therefore, for a completely new dataset that has no correlation with the open-source datasets, we might not expect it to achieve outstanding performance, just as different algorithms always have their suitable application scenarios. **However, we think this issue will not cause a drastic deterioration in our method's performance for the following reasons:**
>
>     * For most tasks, it is generally possible to find tasks that have a certain degree of correlation with them. This correlation can be at the object category level (e.g., tigers and leopards are both large felines) or at the attribute level (e.g., we found that in our dataset, the accuracy of VLM in recognizing the "striped" pattern in the DTD [2] dataset is positively correlated with its accuracy in recognizing "tiger" and "zebra" in Imagenet [3]. This is because the skin patterns of zebras and tigers are striped).
>
>     * For some new datasets that indeed have no correlation with open-source datasets, we can first measure their degree of correlation with the open-source datasets by calculating the text similarity between their class names and those of the open-source datasets. Then, we can set a threshold. Then for open-source tasks with a correlation score lower than the threshold, we do not use our SWAB method, and the method degrades to the original SoTA ModelGPT [1] method. For open-source tasks with a correlation score higher than the threshold, we use our SWAB method to leverage the open-source datasets to help us better select the model.
>
>     * As VLMs are evaluated on an increasing number of tasks, their performance evaluation results on previous tasks are expanded, further enhancing the capability of our method. This creates a virtuous cycle.
>
> [1] Zohar, et al. "Lovm: Language-only vision model selection." NeurIPS 2023.
>
> [2] Cimpoi, Mircea, et al. "Describing textures in the wild." CVPR 2014.
>
> [3] Deng, Jia, et al. "Imagenet: A large-scale hierarchical image database." CVPR 2009.

---

> > ### Comment · Reviewer_mpun · 2024-08-13
> > **Response to Rebuttal**
> >
> > Thank the authors for the Rebuttal. I read the Rebuttal and other reviewers' comments. Since the authors have addressed my concerns, I would like to raise my score to boardline accept.

---

> > > ### Author Response · Authors · 2024-08-14
> > >
> > > Thank you very much for your acknowledgment and efforts.

---

> ### Author Response · Authors · 2024-08-10
>
> Dear Reviewer mpun,
>
> We hope this message finds you well. I am reaching out to kindly request your prompt response to confirm whether our responses adequately address your queries. We sincerely thank you for your time and effort during this discussion period. Your timely feedback is greatly appreciated.

---

> ### Author Response · Authors · 2024-08-12
>
> Dear reviewer mpun,
>
> We thank you sincerely for your time and effort in reviewing our manuscript and providing valuable suggestions. As the author's response period is nearing its conclusion, we kindly remind you to review our response.
>
> We have provided detailed responses to your questions and hope that they adequately address your concerns. If you need further clarification or have any other questions, please feel free to discuss them with us! We are more than willing to continue our communication with you. We would greatly appreciate it if you would update the rating by taking our responses into account.

---

> ### Author Response · Authors · 2024-08-13
>
> Dear Reviewer mpun,
>
> We have responded to your comments in detail. As the discussion period will end in less than one day, we would like to ask whether there are any additional concerns or questions we could address.
>
> Thanks very much for your effort!
>
> Best regards,
>
> Authors

---

### Official Review · Reviewer_omtt · 2024-07-10

**Soundness:** 2
**Presentation:** 3
**Contribution:** 2
**Rating:** 4
**Confidence:** 4

**Summary:**

With the popularity of Vision Language Model (VLM) research in recent years, many versions have emerged, forming the VLM Zoo. This paper aims to select the most appropriate pre-trained VLM from the VLM Zoo, relying solely on texts of the target dataset without access to images. Two challenges are analyzed, namely, “Modality Gap” and “Capability Gap”. One VLM selection method (SWAB) is proposed: first calculate optimal transportation matrix to capture the relevance between open-source and target datasets; then use this matrix to transfer useful statistics of VLMs from open-source to target dataset for bridging two gaps and enhancing the VLM’s capacity estimation. Experiments are carried out across some VLMs and classification datasets to validate the effectiveness.

**Strengths:**

[+] Due to the differences in training data, architecture and pipelines, existing VLMs do have their own strengths in terms of capabilities. Thus, studying how to better utilize members in VLM Zoo under different scenarios is somewhat valuable.

[+] The paper has rich symbols and formalizations, and the space utilization is also reasonable.

[+] Some experiments are conducted to demonstrate the effectiveness of components and designs on a diverse set of datasets of image classification.

**Weaknesses:**

[-] Problem rationality. Although each member in the VLM Zoo has their own strengths, relying solely on textual descriptions to find the most suitable VLM (LOVM) seems too demanding. And language is usually compact, which can easily confuse concepts. Why cannot we use one combination of images and texts (or even text-image pairs) for VLM selection, as images are very easy to obtain in real-world scenarios?

[-] Selection strategy. Given that the members in VLM Zoo have different strengths and abilities, it means that they have somewhat degree of complementarity. One natural idea is to follow ensemble learning, that is, vote on the results of each member, or select the top-k VLMs’ results for complementary combination. Why only choose the most suitable one? Please provide more comparisons and explanations.

[-] Generalization. This paper carries out experiments on many datasets, but all for image classification. Since VLMs excel in handling various tasks, such as detection, segmentation, captioning, and VQA, the generalization on these tasks is still unclear.

**Questions:**

[-] Better visualization. For Fig. 3, there are too many symbols and formulas, increasing difficulty in understanding (the significance of using images is to reduce understanding costs). To this end, the reviewer believes that this paper can be polished better.

**Limitations:**

This paper does not discuss the limitations of the work. However, the reviewer seems to find some limitations, such as the unrealistic assumptions in settings and the ability to only handle classification tasks. Please refer to Weaknesses for more details.

---

> ### Author Rebuttal · Authors · 2024-08-06
>
> We appreciate the reviewer's comments. Below are our responses:
> * Q1: Why not use combination of images and texts (or even text-image pairs) for VLM selection？
> * A1:
>
>     1. Firstly, we want to emphasize that in our paper, "Language-Only" means that for the target new task's dataset, we can only obtain the text descriptions (e.g. category names) of the target task. And we can't obtain the images of the target task. This has been explained in lines 29-31 of our paper. **But for the open-source datasets, we did use the image-text pairs mentioned by the reviewer.**
>
>     2. Secondly, we follow the setting proposed by LOVM [1] in NeurIPS 2023, rather than the setting we devised ourselves. This setting has already been peer-reviewed and validated.
>
>     3. Thirdly, benefiting from image-text pre-training on large-scale data, VLMs represented by CLIP have strong zero-shot capabilities. Therefore, users naturally expect to obtain a model with strong zero-shot capabilities without providing image data. And this aligns with our "Language-Only" setting. This highlights the rationality and practical significance of the setting in our paper.
>
>   4. Finally, we want to emphasize that generating diverse text with LLMs is simpler and more cost-effective than collecting precisely annotated diverse images. To obtain accurate evaluation results, the test image samples need to be sufficiently diverse and accurately labeled. However, collecting such diverse and precisely labeled image samples is quite challenging and may require significant human and financial resources. Images or image-text pairs scraped from the web can be extremely noisy, as pointed out in the CLIP[2], ALBEF[3], and BLIP[4] papers. And as mentioned in the introduction of our paper and by reviewer qHTS, collecting these images is even more difficult for some downstream users, such as non-machine learning practitioners. What they want is to input their task description into a search box, and have a model selection algorithm return the suitable VLMs for their task.
>
>   **Compared to collecting images, generating diverse, high-quality text with LLMs is relatively easy. This opinion is also mentioned in some work[5]. Our paper further analyzes the critical issue of "how to make the diverse text generated by LLMs better substitute for diverse images" and provides a solution, thereby demonstrating significant value and importance of our work.**
>
> * Q2: Why only choose the most suitable VLM?
> * A2:
>
>     1. First, we want to emphasize that our method focuses on model selection, specifically providing an accurate estimate of the performance ranking of VLMs in the VLM Zoo for a target task. This performance ranking estimate helps users make better use of the VLM Zoo for their target tasks, whether selecting the single best model or choosing the Top-K models for an ensemble. In fact, the reviewer's question is concerned with "how to use the VLM Zoo based on the obtained ranking list," while our method focuses on "how to accurately estimate the performance ranking of VLMs in the VLM Zoo for the target task." These are different points of focus.
>
>     2. Secondly, our article prioritizes selecting a single best model mainly because the inference cost of a single model is smaller compared to an ensemble of multiple models, aligning better with the reality that most downstream users lack sufficient computational resources. However, users can still use the performance ranking list provided by our method to select the top-K models for ensemble. We reiterate that the ranking list estimates provided by our method can universally assist various uses of the VLM Zoo, including using a single model or an ensemble.
>
> * Q3: Why only focus on the image classification task?
> * A3:
>
>     1. Firstly, as mentioned in LOVM[1], the reason we do not primarily focus on tasks such as segmentation and object detection is that these tasks usually require the introduction of additional model structures and training processes on top of the original VLM pre-trained model. This leads to additional noise when evaluating VLM performance (for example, using different model fine-tuning methods may result in completely different VLM performance rankings, making it difficult to determine the true ranking of VLMs on the target task). Therefore, our experiments mainly focus on zero-shot image classification tasks.
>
>     2. Secondly, our SWAB is not specifically designed for image classification tasks. For the Modality Gap, our method aims to reduce the distribution differences between different modal data. For the Capacity Gap, our goal is to use the model's ranking in related historical tasks to predict its ranking in new target tasks. These designs are not tied to image classification and can be extended to other zero-shot tasks. For example, in zero-shot image-text retrieval tasks, we can use our SWAB method to better utilize text samples as substitutes for image samples for model evaluation.
>
>     3. Thirdly,  zero-shot image classification tasks are highly representative applications for VLMs and are among the most important and valuable tasks for VLMs. We believe that the excellent model selection results of our method in this task already demonstrate the significance and effectiveness of our approach.
>
>  [1] Zohar et al. Lovm: Language-only vision model selection. NeurIPS 2023.
>
>  [2] Radford et al. Learning transferable visual models from natural language supervision. ICML 2021
>
>  [3] Li et al. Align before fuse: Vision and language representation learning with momentum distillation. NeurIPS 2021.
>
>  [4] Li et al. Blip: Bootstrapping language-image pre-training for unified vision-language understanding and generation. ICML 2022.
>
>  [5] Zhang et al. Diagnosing and Rectifying Vision Models using Language. ICLR 2023.

---

> > ### Comment · Reviewer_omtt · 2024-08-14
> >
> > Thanks to the authors for their efforts during the rebuttal. I have read all the comments from other reviewers and the authors' responses. Some of my concerns have been resolved, such as Q3. Yet, I stand by my opinion that “make the diverse text from LLMs substitute for diverse images” is a bit idealistic, as compact texts cannot cover pixel-level vision. Besides, for Q2, ranking VLMs does not seem to imply the effectiveness of an ensemble. Thus, I decide to slightly raise my score to "borderline reject".

---

> > > ### Author Response · Authors · 2024-08-14
> > >
> > > Thank you very much for your response. We would like to provide the following further clarifications regarding your concerns:
> > >
> > > **Regarding Q1**: We believe that text with different semantics can serve as a substitute for images that match its semantics. In fact, several studies have recognized and validated this perspective. For instance, in [1], the authors state that “collecting a large enough test set that covers different image distributions is a fundamental challenge.” Therefore, they use diverse text generated by large language models as a supplement to image training samples to debias vision models and improve performance (see Figure 1 and Section 2.2 in [1] for more details). Based on our work and the successful implementation of this approach in previous studies, we believe that considering the high noise characteristics of image or image-text pairs collected online, as mentioned in numerous papers [2,3,4], the practice of "using text with different semantics as a substitute for images that match its semantics" is both reasonable and valuable.
> > >
> > > **Regarding Q2**: Firstly, we emphasize that once the model rankings are obtained, these rankings can enhance the performance of an ensemble (e.g., using the top three models most suitable for the current task in an ensemble will generally outperform an ensemble formed by randomly selecting three models from the model zoo). Secondly, as we mentioned in our rebuttal, using multiple models in an ensemble requires significant computational resources. Therefore, in many cases, a common practice to balance performance and efficiency is to select the best single model. As a result, our paper primarily focuses on the scenario of using a single model.
> > >
> > > [1] Zhang et al. Diagnosing and Rectifying Vision Models using Language. ICLR 2023.
> > >
> > > [2] Radford et al. Learning transferable visual models from natural language supervision. ICML 2021
> > >
> > > [3] Li et al. Align before fuse: Vision and language representation learning with momentum distillation. NeurIPS 2021.
> > >
> > > [4] Li et al. Blip: Bootstrapping language-image pre-training for unified vision-language understanding and generation. ICML 2022.

---

> ### Author Response · Authors · 2024-08-10
>
> Dear Reviewer omtt,
>
> We hope this message finds you well. I am reaching out to kindly request your prompt response to confirm whether our responses adequately address your queries. We sincerely thank you for your time and effort during this discussion period. Your timely feedback is greatly appreciated.

---

> ### Author Response · Authors · 2024-08-12
>
> Dear reviewer omtt,
>
> We thank you sincerely for your time and effort in reviewing our manuscript and providing valuable suggestions. As the author's response period is nearing its conclusion, we kindly remind you to review our response.
>
> We have provided detailed responses to your questions and hope that they adequately address your concerns. If you need further clarification or have any other questions, please feel free to discuss them with us! We are more than willing to continue our communication with you. We would greatly appreciate it if you would update the rating by taking our responses into account.

---

> ### Author Response · Authors · 2024-08-13
>
> Dear Reviewer omtt,
>
> We have responded to your comments in detail. As the discussion period will end in less than one day, we would like to ask whether there are any additional concerns or questions we could address.
>
> Thanks very much for your effort!
>
> Best regards,
>
> Authors

---

> ### Author Response · Authors · 2024-08-14
>
> Dear Reviewer omtt,
>
> We have responded to your comments in detail. As the discussion period will end in less than 10 hours, we would like to ask whether there are any additional concerns or questions we could address.
>
> Thanks very much for your effort!
>
> Best regards,
>
> Authors

---

### Official Review · Reviewer_NUg2 · 2024-07-11

**Soundness:** 3
**Presentation:** 3
**Contribution:** 3
**Rating:** 5
**Confidence:** 3

**Summary:**

The paper proposes SWAB the modificaiton of LOVM to mitigate the negative impact of two gaps of LOVM: the modality gap and capability gap. The resutls show the effectiveness of SWAB.

**Strengths:**

1. The motivation of this paper is clear and structure is easy to follow.

**Weaknesses:**

1. For the modality gap, the SWAB using open-source datasets, could you please provide how many image samples used in the experiment. And could you please provide the accuracy vs number of open-source image samples?
2. The SWAB seems like a learnig-based model, could you please add the ablition study for no learning based SWAB.

**Questions:**

1. Why adding Gaussian noise to the target generated embedding?
2. How to select weighted parameter $\alpha$ in Eq.9
3. The Auxiliary texts is generated by ChatGPT, could you please test the open-source LLM like LLama2?

**Limitations:**

No.

---

> ### Author Rebuttal · Authors · 2024-08-05
>
> We appreciate the reviewers' comments. Below are our responses:
> * Q1: How many image samples were used in the experiment.
> * A1: We followed LOVM's approach [1] to use 23 datasets.  Each of the 23 datasets was used as a test dataset in turn, with the remaining 22 datasets serving as open-source datasets. Therefore, the size of the open-source datasets varied for different tasks. The largest open-source dataset contained 497,123 images, the smallest contained 389,080 images, and the average number of images was 476,189.
>
>     It is important to emphasize that the previous SoTA method, ModelGPT [1], also used the same open-source datasets as we did.
>
> * Q2: Provide the accuracy vs number of open-source image samples.
> * A2: We conducted additional experiments by randomly sampling images from the original open-source datasets at different ratios to create new open-source datasets and applied our method based on them. We use the same sampling ratio for each category. We repeated the experiment 5 times and show the averaged the results. We found that the variation among different experimental results was small. The results are as follows:
>     | Data Ratio | 40%   | 60%   | 80%   | 100%  |
>     |------------|-------|-------|-------|-------|
>     | $R_5$       | 0.508 | 0.519 | 0.525 | 0.534 |
>     | $\tau$       | 0.247 | 0.256 | 0.253 | 0.260 |
>     | $R_5$ + $\tau$  | 0.755 | 0.775 | 0.778 | 0.794 |
>
>     The result shows that our method is robust to the number of data in the open-source datasets used. This is because the VLM's average gap vector of each category and  the VLM's ranking of each category are robust to data volume. As shown in Table 5 in our paper, gap vectors for different samples within the same category are consistent. Additionally, reducing the dataset size has minimal impact on model rankings.
>
> * Q3: Add the ablation study for no learning based SWAB.
> * A3: Firstly, we emphasize that our method focuses on bridging the modality gap and capacity gap in VLM selection. In achieving these goals, we do not introduce any learning process. In SWAB, only the Rank Model in the third step depicted in Figure 3 of our paper requires training. And this Rank Model is introduced by the ModelGPT we follow. We use our  SWAB to optimize the ModelGPT's capabilities by bridging the modality gap.
>
>     Regarding the reviewers' concerns about the improvements of our method over non-learning based methods, the corresponding results are shown in Table 3 of our paper's main text and Figure 5 in the appendix.  The results shown in both table 3 and figure 5 do not use the rank model, and therefore, no training process is introduced:
>     * In Table 3 of the main text, We demonstrated the performance improvement in selecting the VLM by using the weighted average of VLM's ranking on categories of open-source dataset with SWAB to bridge the capacity gap, compared to using the average VLM's ranking of various categories of open-source data directly.
>     *  In Figure 5 of the appendix, we separately show the top-1 classification accuracy calculated on text samples before and after using SWAB to bridge the modality gap between text and image data. This demonstrates a better rank consistency between the text top-1 classification accuracy of VLMs and actual classification accuracy of VLMs after using our SWAB.
>
> * Q4: Why adding Gaussian noise to the target generated embedding?
> * A4: We added Gaussian noise to simulate the variance in image samples because different images can correspond to the same caption. For example, in the text "A dog running on the grass," the dog in the images can vary in breed. We add Gaussian noise to the features of text samples to simulate the diversity of images. We found that adding Gaussian noise improves the performance of our method. Some existing works [1, 2] have also adopted similar approaches.
>
> * Q5: How to select weighted parameter in Eq.9.
> * A5: $\alpha$ is a hyperparameter. We used the same alpha value of 0.45 for all experiments.
>
> * Q6: Test the open-source LLM like Llama2 to generate the Auxiliary texts.
> * A6: We used Llama 2 to generate the corresponding text and found that our method performs excellently with texts generated by different LLMs. We attribute this to the fact that generating category-specific diverse texts is a relatively simple task for LLMs, and common LLMs possess this capability.
> | Metrics | $R_5$  | $\tau$ | $R_5 + \tau$ |
> |---------|------|------|----------|
> | Llama2  | 0.527| 0.254| 0.781    |
>
> [1] Zohar, Orr, et al. "Lovm: Language-only vision model selection." NeurIPS 2023.
>
> [2] Zhang, Yuhui, et al. "Connect, Collapse, Corrupt: Learning Cross-Modal Tasks with Uni-Modal Data." ICLR 2024.

---

> ### Author Response · Authors · 2024-08-10
>
> Dear Reviewer NUg2,
>
> We hope this message finds you well. I am reaching out to kindly request your prompt response to confirm whether our responses adequately address your queries. We sincerely thank you for your time and effort during this discussion period. Your timely feedback is greatly appreciated.

---

> ### Comment · Reviewer_NUg2 · 2024-08-12
>
> Thank you for your detailed response and for addressing my concerns. However, I have a few more questions. Is the parameter
> $\alpha$ sensitive to different datasets, and what is the best strategy for selecting it? Additionally, could you provide some analysis on this? Another question for clarification: Regarding the experiment setting in Q2, is the performance measured on just one specific test dataset, or is it an average across all 23 test datasets?

---

> > ### Author Response · Authors · 2024-08-12
> >
> > We appreciate your response and are happy to address your issue. Here is our reply:
> >
> > * Q1: Is the parameter $\alpha$ sensitive to different datasets, and what is the best strategy for selecting it? Provide some analysis on this.
> > * A1: Firstly, we emphasize that we use the same $\alpha$ across all datasets. The reasons for this are: (1) we found that using the same $\alpha$ achieves good performance on most datasets and (2) applying the same $\alpha$ to all datasets is a simple and effective approach.
> >
> >     Regarding the strategy for selecting $\alpha$, our experiments revealed that our method's performance is fairly robust to different $\alpha$ values, so we chose an $\alpha$ that performs well on our benchmark. To provide relevant analysis on this point, we varied the $\alpha$ values and demonstrated our method's performance metrics on the LOVM Benchmark.
> >
> >     | $\alpha$   | 0.2   | 0.3   | 0.4   | 0.5   | 0.6   | 0.7   | 0.8   | ModelGPT |
> >     |---------|-------|-------|-------|-------|-------|-------|-------| -------|
> >     | $R_5$     | 0.487 | 0.513 | 0.478 | 0.470 | 0.452 | 0.461 | 0.487 | 0.457 |
> >     | $\tau$     | 0.301 | 0.206 | 0.264 | 0.296 | 0.330 | 0.290 | 0.264 | 0.197 |
> >     | $R_5+\tau$   | 0.788 | 0.719 | 0.742 | 0.766 | 0.782 | 0.751 | 0.751 | 0.654 |
> >
> >     The experimental results show that, for different values of $\alpha$, our approach consistently outperforms the current SoTA method, ModelGPT, demonstrating the robustness of our method to the choice of $\alpha$.
> >
> > * Q2: Regarding the experiment setting in Q2, is the performance measured on just one specific test dataset, or is it an average across all 23 test datasets?
> > * A2: It is an average across all 23 test datasets.

---

> ### Author Response · Authors · 2024-08-13
>
> Dear Reviewer NUg2,
>
> We have responded to your comments in detail. As the discussion period will end in less than one day, we would like to ask whether there are any additional concerns or questions we could address.
>
> Thanks very much for your effort!
>
> Best regards,
>
> Authors

---

> > ### Comment · Reviewer_NUg2 · 2024-08-13
> >
> > Thanks for your response and addressing my concerns. I will keep my score.

---

> > > ### Author Response · Authors · 2024-08-14
> > >
> > > Thank you very much for your acknowledgment and efforts.

---

### Official Review · Reviewer_qHTS · 2024-07-12

**Soundness:** 3
**Presentation:** 3
**Contribution:** 3
**Rating:** 7
**Confidence:** 5

**Summary:**

This article focuses on selecting the best model from a visual-language multimodal model zoo for specific downstream tasks without having images of those tasks. It provides a detailed analysis of two challenges faced in this problem — Modality Gap and Capacity Gap. This paper proposes a method called SWAB, which leverages the statistical measures of VLM on open-source tasks based on the category similarity between historical open-source tasks and target tasks, to mitigate the negative impacts of these gaps on Language-Only VLM model selection. Experimental results demonstrate the effectiveness of the proposed method.

**Strengths:**

1.The problem studied in this paper is novel and interesting, with significant practical value. In recent years, the number of open-source VLMs has been increasing, and their zero-shot capabilities have been improving. For practical downstream users, especially those who lack computational resources, have limited funding to label extensive evaluation datasets, or are non-machine learning practitioners with little experience in setting up and evaluating various VLMs, the Language-Only Vision Model selection (LOVM) problem studied in this article is important.
2.This paper presents a clear and well-motivated problem statement. The authors provide a detailed analysis of the negative impacts of the Modality Gap and Capacity Gap on Language-Only Vision Model (LOVM) selection through explanations and experimental validation. They demonstrate that we cannot directly use text samples related to categories in the target dataset as substitutes for test image samples, nor can we use the average performance of VLMs on open-source datasets as proxies for their performance on specific tasks. This problem statement is compelling and well-supported.
3.The method presented in the paper is novel and interesting. It leverages the category similarity between target tasks and open-source tasks to estimate the corresponding gap vectors and accuracy/ranking of VLMs on the target tasks, based on their gap vectors and accuracies/rankings on the open-source tasks. This idea is grounded in a reasonable assumption: the statistics of VLMs on similar categories are likely to be similar. By doing so, the method effectively utilizes the rich information contained in category similarity to make targeted use of the models' statistics on historical tasks.
4.The evaluation approach in the article is comprehensive and detailed. The benchmark includes 23 commonly used image classification datasets and 43 widely recognized VLMs (such as CLIP, CoCa, BLIP, BEiT, among others, with different pre-training methods and model architectures), making the experimental results highly convincing. Additionally, the article compares its method with various baseline approaches, including the current state-of-the-art method ModelGPT, and demonstrates its significant superiority over these methods. The article also conducts thorough ablation studies to validate the effectiveness of each component of the proposed method.

**Weaknesses:**

1.The performance of ModelGPT is slightly inconsistent with the results in its paper. What are the reasons for this？
2.The method of setting the \alpha value in formula 9 is not stated. Is the same \alpha used for all data sets? What is the exact value?

**Questions:**

Refer to the weakness.

---

> ### Author Rebuttal · Authors · 2024-08-05
>
> We greatly appreciate the reviewers' recognition of our work. Below are our responses to the relevant questions:
> * Q1: The performance of ModelGPT is slightly inconsistent with the results in its paper.
> * A1: This is because we conducted multiple repeated experiments using different random seeds. ModelGPT added Gaussian noise to perturb the features of the text samples. To enhance the reliability of the results, we used 10 different random seeds for repeated experiments and reported the mean and standard deviation of the results. Since the original ModelGPT paper did not perform repeated experiments, our results show slight differences compared to their paper.
>
> * Q2: The method of setting the $\alpha$ value in formula 9.
> * A2: $\alpha$ is a hyperparameter. We used the same alpha value of 0.45 for all experiments.

---

> ### Author Response · Authors · 2024-08-10
>
> Dear Reviewer qHTS,
>
> We hope this message finds you well. I am reaching out to kindly request your prompt response to confirm whether our responses adequately address your queries. We sincerely thank you for your time and effort during this discussion period. Your timely feedback is greatly appreciated.

---

> > ### Comment · Reviewer_qHTS · 2024-08-11
> >
> > Thanks for your response and my concerns are solved.

---

> > > ### Comment · Area_Chair_Yvdz · 2024-08-14
> > > **Impact of the new members in the model zoo**
> > >
> > > Another difference might stem from adding new members in the Model Zoo. Could you quantify the impact of this? For example, what is the performance of your approach using same Model Zoo in the LOVM paper?

---

> > > > ### Author Response · Authors · 2024-08-14
> > > >
> > > > Dear Area Chair,
> > > >
> > > > Thank you very much for your attention to our work. In fact, we initially conducted our experiments on the 35 models provided in the LOVM Benchmark. However, we later received some suggestions from other researchers to include a greater variety of vision-language models (e.g. BLIP and BEIT-3) to enhance the diversity of models in the benchmark, thereby increasing the credibility of the evaluation results. Therefore, we expanded the variety of models based on the LOVM Benchmark. To address your concerns, we are also providing our experimental results on the original 35 models from the benchmark.
> > > >
> > > > | Method                | ModelGPT | SWAB (Ours) |
> > > > |-----------------------|----------|-------------|
> > > > | $R_5$                  | 0.452    | **0.501**       |
> > > > | $\tau$                   | 0.201    | **0.291**       |
> > > > | $R_5 + \tau$            | 0.653    | **0.792**       |

---

> > > > > ### Comment · Area_Chair_Yvdz · 2024-08-14
> > > > > **Thanks, but inconsistent numbers**
> > > > >
> > > > > Thanks for the quick response. Adding new models makes sense, but makes the comparison harder. Also ModelGPT reports:
> > > > > INB+G+C 0.461 (R5) 0.223 (\tau) for top-1 accuracy according to Table 2 in their paper, which are a fair bit higher than your table above. Could you clarify how the results were obtained?

---

> > > > > > ### Author Response · Authors · 2024-08-14
> > > > > >
> > > > > > Dear Area Chair,
> > > > > >
> > > > > > Thank you very much for your interest in our work. **First, we want to emphasize that we used the code provided by LOVM to ensure consistency with the evaluation methods of their benchmark.** We think the slight differences between the ModelGPT results we reported and those in the LOVM paper can be attributed to two main reasons:
> > > > > >
> > > > > > 1. **The use of different random seeds.** As we mentioned in our response to Reviewer NUg2's Q4, in both the LOVM paper and ours, we add Gaussian noise to the text features to simulate the variance in image samples because different images can correspond to the same caption. This process introduces randomness. We noticed that in the code provided by LOVM, no random seed was specified, and they also did not report the random seed used in their experiments. To ensure the reliability of our experimental results, we repeated the experiments using 10 different random seeds (from 1 to 10) and reported the mean and standard deviation of the results. Therefore, the difference in random seeds may lead to slight variations between our results and theirs.
> > > > > >
> > > > > > 2. **The text generated by the LLM used may differ.** In the GitHub repository provided by LOVM, the specific text collection generated by ChatGPT was not directly provided; instead, they included a Python script to generate the text collection using ChatGPT. In our experiments, we used this provided Python script to generate the text collection with ChatGPT. Considering the inherent randomness in LLM-generated responses, the text collection we used might differ slightly from the one used by LOVM. This variability in LLM-generated responses can also contribute to the slight differences between our results and theirs.
> > > > > >
> > > > > > We emphasize that in our experiments, we used the same evaluation methods as LOVM. Furthermore, when comparing ModelGPT with our approach, we ensured that the data and random seeds were completely consistent to guarantee the comparability of the experimental results.
> > > > > >
> > > > > > Thank you very much for your attention and suggestions regarding our work. We hope our responses address your concerns.

---

> > > > > > > ### Comment · Area_Chair_Yvdz · 2024-08-14
> > > > > > >
> > > > > > > Thank you again for the quick responses. The reasons you provided make sense as potential sources of discrepancy between the results. It is unfortunate that the authors of LOVM have not released a reproducible version of their code. However, please note that with your implementation, ModelGPT performs no better than the ImageNet baseline according to the R5 metric.

---

> > > > > > > > ### Author Response · Authors · 2024-08-14
> > > > > > > >
> > > > > > > > Dear Area Chair,
> > > > > > > >
> > > > > > > > Thank you for your attention to our work. We think that ModelGPT's $R_5$ could indeed be lower than ImageNet. This is because, within LOVM, it utilizes the actual performance of various models on open-source datasets as a training set's label to train a Linear Regression Model  to predict performance or ranking of the VLMs (for details, see Section 3 of LOVM). The inputs of this Linear Regression Model include the current model's score on the ImageNet Baseline and some performance metrics calculated on text samples generated by LLM (such as text-acc1 and text-f1, details in Section 3.2 of LOVM). Therefore, **the performance prediction of the Linear Regression Model for the current model actually depends on the weighted sum of scores on the ImageNet Baseline and the performance metrics calculated on text samples produced by LLM.**
> > > > > > > >
> > > > > > > > However, as analyzed in our paper regarding the Modality Gap, **due to the existence of a modality gap between image features and text features in the feature spaces of many VLMs, text samples cannot directly serve as a good substitute for image samples**. Therefore, **we cannot expect that the performance metrics calculated directly on text samples generated by LLMs (such as text-acc1 and text-f1) will positively correlate with the actual performance of the current model,** which in turn affects the performance of the linear regression model in LOVM that uses metrics like text-acc1 as inputs. We believe there are two pieces of evidence that can substantiate this idea:
> > > > > > > >
> > > > > > > > 1. In Figure 5 of our paper, we analyzed the consistency between the text top-1 accuracy calculated on text samples and the actual accuracy of the model before and after using our method to bridge the modality gap. The experimental results show that before bridging the modality gap, the consistency between text top-1 accuracy and the model's actual accuracy was poor. However, **our method significantly improves the consistency between the two by bridging the modality gap, demonstrating the effectiveness of our approach.**
> > > > > > > >
> > > > > > > > 2. We observed that in the original paper of LOVM, the improvement in the $R_5$ metric when using ModelGPT compared to the ImageNet Baseline is very slight (0.461 vs 0.452). **This modest increase could likely disappear due to the randomness of the methods mentioned in our previous response** (i.e., this improvement might fall within the error bars, so after changing to a new random seed, this improvement might disappear). This indicates that the introduction of the performance metrics calculated directly on text samples generated by LLMs provides minimal help.
> > > > > > > >
> > > > > > > > Thank you very much for your attention and suggestions regarding our work. We hope our responses address your concerns.

---

> > > ### Author Response · Authors · 2024-08-14
> > >
> > > Thank you very much for your acknowledgment and efforts.

---

### Author Rebuttal · Authors · 2024-08-07

We would like to express our deepest gratitude to the reviewers for the meticulous examination of the paper and their insightful and valuable comments. We acknowledge that most reviewers observed the shining point, saying the proposed method is novel (qHTS, mpun), and interesting (qHTS, mpun), well-motivated (qHTS,NUg2,omtt,mpun), effective(qHTS,mpun). They also think our experiment results is comprehensive to indicate good validation of the proposed method(qHTS,mpun), and the question we studied is valuable and important (qHTS,omtt,mpun).

Please check the answer to specific comments. Due to the rebuttal limitation, each response is limited to 6000 characters. **We have to control the response content due to this limitation, but we are more than happy to discuss this paper and any further questions with all reviewers in the next week.**

---

### Decision · Program_Chairs · 2024-09-25

**Decision:**

Accept (poster)

**Comment:**

The paper tackles the problem of selecting a visual language model (VLM) from a ModelZoo for a target task using only the labels of the dataset. Two challenges are addressed in the language-only VLM selection (LOVM) problem: the modality gap and the capacity gap. The modality gap refers to the difference between the label (or text) domain and the full task, while the capacity gap refers to the generalization ability of the selector across datasets. Results for model selection are presented on a wide range of datasets and are compared with baselines and prior work, such as ModelGPT.

The reviews for the paper are mostly positive (Accept, Borderline Accept, Borderline Reject, and Borderline Accept).

Reviewer qHTS finds the problem to be novel and interesting with practical value. However, they note that the comparisons with ModelGPT are inconsistent. The rebuttal notes that this might be due to differences in random seeds and other experimental details not described in the paper or the code from the authors. Another difference is due to a larger model zoo. This was clarified in the rebuttal, where results using an identical model zoo from the ModelGPT paper were shown, and the method outperformed the baselines. However, the differences in the numbers seemed fairly large despite the authors' best efforts.

Reviewer NUg2 found the paper well-motivated and had questions regarding some experimental settings. However, their final score remained unchanged after the rebuttal, with a borderline accept rating.

Reviewer omtt was the most critical and questioned the applicability of text-only model selection. They also raised the question of how the model might handle tasks beyond classification. While the rebuttal addressed some concerns, the final rating was a borderline reject.

Reviewer mpun found the paper to be novel and noted that this is the first time LOVM has been applied. However, the Area Chair (AC) notes that LOVM has been proposed in prior work. The reviewer also found the approach well-motivated, and the experiments to be extensive. The main weaknesses were addressed in the rebuttal, and the score was raised to a borderline accept.

The AC read the paper, reviews, rebuttal, and discussions, and generally agrees with the positive sentiment about the paper. However, there are some issues the authors should address in the final version. The first is to make the comparison with ModelGPT easier. Perhaps the authors could report results using the identical model zoo and present a standalone evaluation with identical text. These experimental differences should be documented in the paper to facilitate future comparisons. Second, the AC found the related work section of the paper to be lacking. Some works, such as Task2Vec, are not cited, where the terms domain2vec and label2vec are introduced. Label2vec presents a way to represent tasks using only some statistics of the labels, which is related to this paper.

Task2Vec: Task Embedding for Meta-Learning, Achille et al., ICCV’15